# Wake slow waves in focal human epilepsy impact network activity and cognition

Laurent Sheybani [1,2,3], Umesh Vivekananda[1,2,3], Roman Rodionov[1,2,3], Beate Diehl[1,2,3], Fahmida A. Chowdhury[1,2,3], Andrew W. McEvoy[1,2,3], Anna Miserocchi[1,2,3], James A. Bisby [4], Daniel Bush [5] ✉, Neil Burgess [1,6] ✉ & Matthew C. Walker [1,2,3] ✉

Slow waves of neuronal activity are a fundamental component of sleep that are proposed to have homeostatic and restorative functions. Despite this, their interaction with pathology is unclear and there is only indirect evidence of their presence during wakefulness. Using intracortical recordings from the temporal lobe of 25 patients with epilepsy, we demonstrate the existence of local wake slow waves (LoWS) with key features of sleep slow waves, including a down-state of neuronal firing. Consistent with a reduction in neuronal activity, LoWS were associated with slowed cognitive processing. However, we also found that LoWS showed signatures of a homeostatic relationship with interictal epileptiform discharges (IEDs): exhibiting progressive adaptation during the build-up of network excitability before an IED and reducing the impact of subsequent IEDs on network excitability. We therefore propose an epilepsy homeostasis hypothesis: that slow waves in epilepsy reduce aberrant activity at the price of transient cognitive impairment.

During non-rapid eye movement (NREM) sleep, neurons undergo slow fluctuations of membrane potential, alternating between a burst firing mode (up-state) and a suppression of their activity (down-state). This fluctuation is reflected by slow oscillations (0.5–4 Hz) of the local field potential (LFP) termed slow wave activity (SWA, Fig. 1a). SWA is a canonical component of sleep[1,2] and has been proposed to be critical for sleep homeostasis[3–7] (specifically, the normalization of synaptic strength[6,8]), metabolic regulation[9], and metabolic waste clearance in general[10]. Elevated sleep pressure, as occurs after a wake period, is reflected by an increased rate, slope and amplitude of sleep slow waves (SWs), which decrease over the course of the night[4,11–15] (Fig. 1b). Furthermore, evidence suggests that neuronal synchrony – e.g. increased neuronal firing during the up-state and neuronal silence during the down-state of sleep SWs – favors synaptic normalization[8,11,16] and is homeostatically modulated (high after a wake period, low after a sleep

period, Fig. 1a)[12]. Importantly, such slow wave activity is distinct from the focal slowing (i.e., intermittent or persistent theta/delta frequency oscillations) that occurs in pathological circumstances such as epilepsy, and presumably reflects different generative mechanisms[17].

Like wakefulness[4] and cognitive processing[3,18], epileptic activity results in increased synaptic connectivity[19–21] and metabolic need[22]. Consequently, the increased SWA observed during sleep after repetitive seizures has been interpreted as a compensatory mechanism[17,23] which counteracts the increased local metabolic demand. However, this contrasts with the view that SWs occurring in pathological brain regions are detrimental[24,25] or even pro-epileptic[26] and there is currently no evidence for SW having a beneficial impact on epileptic activity. Furthermore, it is unclear whether such SWs might also appear during wakefulness, although this could constitute a mechanism to offset pathological activity, such as epileptiform discharges and seizures.

[1]Department of Clinical and Experimental Epilepsy, UCL Queen Square Institute of Neurology, University College London, London, UK. [2]National Hospital for Neurology and Neurosurgery, University College London Hospitals NHS Foundation Trust, London, UK. [3]NIHR University College London Hospitals Biomedical Research Centre, London, UK. [4]Division of Psychiatry, University College London, London, UK. [5]Department of Neuroscience, Physiology and Pharmacology, University College London, London, UK. [6]Institute of Cognitive Neuroscience, University College London, London, UK. ✉e-mail: drdanielbush@gmail.com; n.burgess@ucl.ac.uk; m.walker@ucl.ac.uk

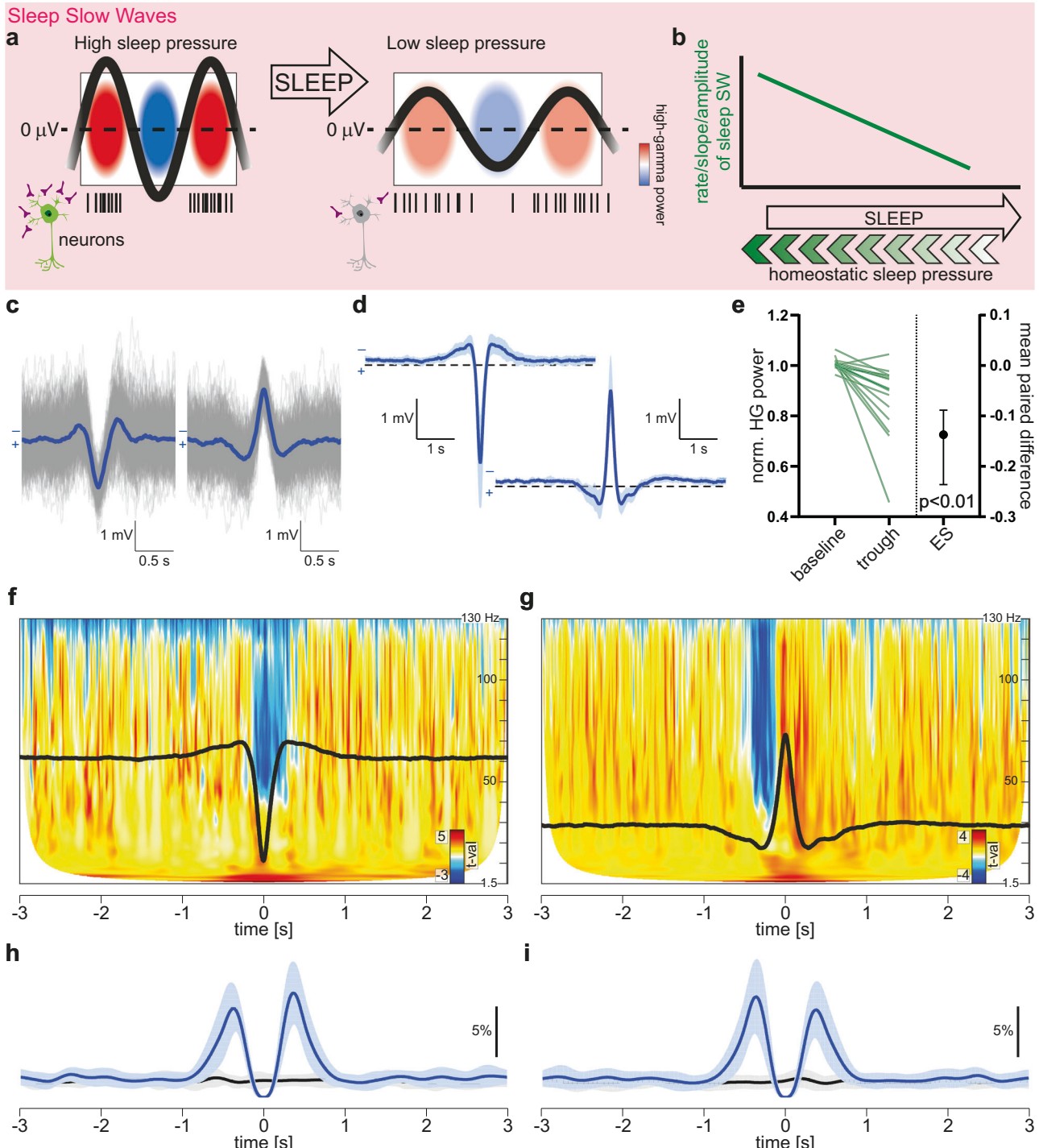

**Fig. 1 | Local wake slow waves (LoWS) recapitulate defining features of sleep slow waves. a**, **b** Properties of sleep slow waves (SWs). After a period of wake-fulness, SWs present with increased rate, slope and amplitude, reflecting increased homeostatic sleep pressure. Neuronal activity (black vertical lines) aligns with the up-state of SWs. High-gamma (HG, 45–130 Hz) power, color-coded from blue (low) to red (high), reflects this increased neuronal activity. During the down-state, neurons are silent (low HG power). Synapses (purple) are pruned and neurons move from more (green) to less (gray) excitable. **b** After a period of sleep, the rate, slope and amplitude of SWs decrease, reflecting the decrease in homeostatic sleep pressure. **c** Example of identified negative and positive slow waves during wakefulness (nSW and pSW respectively, patient 3, electrode 1, 247 nSW and 207 pSW, mean in blue). nSW and pSW correspond to positive and negative extracellular polarity respectively, the former reflecting hyperpolariza-tion of neurons, a core feature of sleep SW. **d** Grand average LoWS across

participants (mean ± SD). **e** Normalized HG (45–130 Hz) power during the nSW-associated down-state (averaged between the two 0-crossings of nSW) shows a significant decrease relative to baseline (two-sided permutation paired $t$-test, $p = 0.0004$, $n = 17$ patients, ES: mean and 95% CI). Time-frequency decomposition of nSW (**f**) and pSW (**g**) converted into t-statistic values. Black line: grand average nSW and pSW. Note the drop in HG power at nSW trough. In contrast, the drop in HG power precedes pSW and corresponds to the peak of nSW occurrence before pSW. Panels g and i are temporally aligned. Mean ± SD ($n = 17$) co-occurrence of pSW and nSW (**h**) and the inverse (**i**), using true position (blue) and shuffled position (black, control) of nSW and pSW. nSW and pSW thus occur mainly as 3 peaks (1.5 oscillatory cycles). Note the alignment of the peak of nSW occurrence before pSW and the drop of HG power seen in **g**. Scale: per patient, per nSW and pSW, per unit of time (i.e., percentage of nSW and pSW that is preceded/followed by another wave). ES effect size.

Here, using intracranial macro- and micro- electrode recordings from the temporal lobe of people with focal, pharmacoresistant epilepsy, we demonstrate the presence of highly local wake slow waves (LoWS) that recapitulate the core features of sleep slow waves, including the associated down-state of neuronal spiking activity. Importantly, LoWS are isolated, discrete events that do not correlate with overall delta power, indicating that they are distinct from the focal slowing / delta oscillations typically associated with brain lesions and epilepsy in particular[27]. We hypothesized that these LoWS could serve a homeostatic purpose by normalizing neuronal activity to prevent epileptic discharges, mirroring the function of sleep SWs which correct the excessive neuronal excitability accumulated during wakefulness that translates into high sleep pressure (i.e., sleep homeostasis[6]). If LoWS do serve a homeostatic function, then they should exhibit two key features: first, responding to increases in network excitability that precede interictal epileptiform discharges (IEDs); and second, reducing abnormal activity linked with IEDs. In line with this hypothesis, we observed that progressive increases in neuronal activity (estimated by high-gamma (HG) power) before IEDs are accompanied by an increase of the slope and amplitude of LoWS, analogous to the response of sleep SW to increased sleep pressure[13,15]. Moreover, we found that a longer delay since the last LoWS is associated with higher HG power during IEDs, suggesting that any protective function of LoWS dissipates with time. Lastly, we found that a higher rate of LoWS during an associative memory task was associated with longer reaction times (RT), supporting the prediction[5] that the substantial modulation of neuronal activity during SWs[12,28] impacts cognitive processing. Together, our findings indicate that temporal lobe LoWS with key features of sleep SW dynamically respond to changes in network excitability, reduce aberrant activity associated with IEDs, and impact on cognition. We therefore propose that LoWS represent a homeostatic process that comes at the cost of transient cognitive impairment.

## Results

### Slow waves during wakefulness in patients with epilepsy

We first asked if we could reliably detect local slow waves during wakefulness in a cohort of people with focal epilepsy ($n = 17$) undergoing intracranial EEG recordings, and then whether such slow waves had key features of sleep slow waves (Fig. 1). We used a validated algorithm for slow wave detection[13,26,28,29] to track their presence in the human hippocampus (mean ± SD hippocampal electrode contacts per patient = 3.4 ± 1.8, see Supplementary Table 1), while people with focal epilepsy were performing an associative memory task. We excluded any slow waves associated with (i.e., within 1 s following) an IED to ensure that identified events were not post-IED slow waves and also took several confirmatory steps to ensure that the identified slow waves were not mislabeled IEDs (see Online Methods).

We identified typical SWs with a positive polarity (i.e., positive extra-cellular potential but negative EEG signature, nSW) and with an incidence rate of $0.06 \pm 0.02$ s$^{-1}$ electrode$^{-1}$ during wakefulness (Fig. 1c-d). Consistent with this, we also found wake nSW in another dataset from patients with focal epilepsy undergoing intracranial recordings, although at a lower rate (from Boran et al.[30], referred to as the "Boran dataset" below, Boran vs UCL dataset: effect size [ES] expressed as mean difference, 95% CI: 0.0294, 0.0164−0.0398 s$^{-1}$ electrode$^{-1}$, $p = 0.0004$, $n = 9$ and 17 patients respectively, Supplementary Fig. 1a, b). Finally, we also found wake SW in a third independent dataset[31–33] (from the MNI Open iEEG Atlas, referred to as the "MNI dataset" below, Supplementary Fig. 1c), but could not clearly distinguish nSW from pSW due to the bipolar montage used (see Methods). Nonetheless, the MNI dataset provided an opportunity to compare the incidence of SWs across wake-sleep stages. This confirmed that SWs were rarer during wakefulness than during sleep (see Supplementary Fig. 1d). The lower rate during NREM 2 probably reflects the fact that, by definition[1], a maximum of 20% of slow-wave

activity can be detected in any 30 s window, while NREM 3 can be comprised of 20–100% of slow-wave activity. Furthermore, since the MNI dataset includes presumably healthy brain regions, the finding of wake SWs in these regions indicates that they were not strictly restricted to epileptic brain regions.

We next asked whether these isolated nSW simply reflected the background slowing/increased delta activity typically associated with epileptic foci or increased delta power that dominates hippocampal activity during wakefulness[31]. First, we observed that nSW were, qualitatively, isolated, discrete events, and not continuous oscillations (Supplementary Fig. 2). Furthermore, their incidence rate was not correlated with delta or theta power across the whole recording (Supplementary Fig. 3a, b) and the delta-band oscillatory content around nSW and during control periods was similar (ES, 95% CI: 0.002, −0.004 to 0.007 a.u., $p = 0.501$, $n = 17$, Supplementary Fig. 3c−e; oscillatory content: see Methods for details), indicating that they did not occur during transient oscillatory activity. Finally, we compared the incidence rate of nSW on mesial temporal lobe electrodes (MTL, amygdala and hippocampus) between patients with mesial ($n = 6$) and extra-mesial temporal ($n = 10$; excluding one patient in which the seizure onset zone could not be localized) lobe epilepsy and did not find a significant difference (ES, 95% CI: 0.05, −0.02 to 0.14 s$^{-1}$, $p = 0.226$, Supplementary Fig. 4). Altogether, this indicates that nSW were distinct from the pathological slowing typically observed in epileptogenic regions and their rate was not increased in the epileptic focus, at least in the case of mesial temporal lobe epilepsy.

Next, we isolated the negative and positive EEG signatures of slow waves, i.e., negative and positive slow waves (nSW and pSW respectively, Fig. 1c, d). Rates of nSW and pSW were similar (ES, 95% CI: 0.002, −0.001 to 0.009 s$^{-1}$ electrode$^{-1}$, $p = 0.464$, $n = 17$ patients, Supplementary Fig. 5a) and highly correlated across patients (Pearson correlation, $r^2 = 0.79$, $p < 0.0001$, $n = 17$, Supplementary Fig. 5b). nSW had a peak (extracellular) positive potential (Fig. 1c, d), which corresponds to a hyperpolarization of neurons. This inactive state is a core feature of, and defines, sleep SW[5,28]. For this reason, we subsequently focus only on the negative component of this slow wave activity (i.e., nSW), except where stated otherwise. The median oscillatory frequency of nSW, estimated by the delay between the two 0-crossings, was 1.3 Hz (SD: 0.08 Hz, $n = 17$ patients).

### Wake slow waves share key characteristics with sleep slow waves

Next, we determined whether identified slow waves had key characteristics of sleep SW (Fig. 1). Analogous to sleep SW[28], nSW were associated with a down-state, as revealed by decreased high-gamma (HG, 45–130 Hz) power averaged between the two 0-crossings of the grand average nSW (−0.15 to 0.165 s around the trough of nSW versus baseline: ES, 95% CI: −0.137, −0.236 to −0.0884 a.u., $p = 0.0004$, $n = 17$, Fig. 1e). Using a time-resolved frequency analysis, we confirmed that the decrease in HG power was localized to the nSW trough (Fig. 1f). This was further confirmed by the observation of a decrease in HG power before the peak of pSW (from −0.42 to −0.13 s, Fig. 1g), the timing of which matched the peak of nSW occurrence before pSW in our data (around −0.36 s, Fig. 1g, i).

In the time domain, up to 9–10% of nSW were preceded or followed by a pSW (with a delay of ± 0.37 s, Fig. 1h). The same analysis for pSW gave, as expected, a similar result (Fig. 1i). This rate of temporal coincidence was significantly above the chance rate (nSW around pSW: significant from −0.95 s to 0.05 s, then from 0.25 s to 0.95 s; pSW around around nSW: from −0.85 s to 0.05 s, then from 0.25 s to 0.85 s, $n = 17$, see Methods and black traces in Fig. 1h−i), suggesting that nSW and pSW typically occurred as short trains of 2-3 half-waves. In the spatial domain, the presence of a slow wave on one hippocampal contact was accompanied by a slow wave on only 27% of the other hippocampal contacts (Supplementary Fig. 6a). This microfocality was less pronounced in the amygdala (49%, Supplementary Fig. 6b) and

more pronounced in the temporal neocortex (18%, Supplementary Fig. 6c), which could reflect differences in the volume and coverage of these brain regions. Similarly, the detection of a slow wave in the hippocampus, amygdala or temporal neocortex was accompanied, on average, by the detection of a slow wave in at least one of the other regions in 24% of cases (Supplementary Fig. 6d). Slow wave incidence rates were similar in the hippocampus (mean ± SD: $0.06 \pm 0.02 \, s^{-1}$ electrode$^{-1}$, $n = 17$), amygdala ($0.06 \pm 0.01 \, s^{-1}$ electrode$^{-1}$, $n = 12$), and temporal neocortex ($0.06 \pm 0.02 \, s^{-1}$ electrode$^{-1}$, $n = 17$, Supplementary Fig. 6e). Given the highly focal nature of detected nSW, we subsequently refer to them as local wake slow waves (LoWS).

## LoWS are associated with a down-state of neuronal firing

To confirm that the decrease in HG power during the LoWS trough corresponds to decreased neuronal activity, we examined LoWS in a separate cohort of patients who underwent hippocampal microelectrode implantation ($n = 8$ patients) and identified unit spiking activity (see Methods and Fig. 2a). We observed a significant decrease in unit activity locked to the trough of LoWS (paired bootstrap statistic followed by false discovery rate correction across time bins, $p < 0.05$ from −0.049 to 0.147 s, Fig. 2b). The maximal decrease of neuronal firing occurred just after the trough (Fig. 2a–d), as previously described for

sleep SWs in the hippocampus[28]. At this time point, >50% of identified units in all 8 patients showed a decrease in firing rate (mean ± SD = 78% ±17%, one sample $t$-test, $p = 0.0024$, Fig. 2c, $n = 8$). Similarly, a comparison of normalized mean firing rates between the two 0-crossings of the grand average LoWS versus baseline confirmed that the decrease was significant (ES, 95% CI: −0.273, −0.455 to −0.0795, $p = 0.0368$, $n = 8$, Fig. 2e blue lines). Conversely, we found only non-significant trends in the modulation of neuronal activity around pSW (red lines in Fig. 2e) and IEDs (Supplementary Fig. 7a, b). In sum, neuronal activity shows a significant decrease shortly after the trough of LoWS during wakefulness, confirming the presence of this core feature of sleep slow waves in LoWS.

## LoWS and interictal epileptiform discharges

Next, we asked if there was any interaction between LoWS and interictal epileptiform discharges (IEDs). IEDs were identified in all but one patient (mean rate ±SD across patients and electrodes: $0.01 \pm 0.01 \, s^{-1}$ electrode$^{-1}$, $n = 16$, Fig. 3a–c). First, we examined whether the incidence rate of IEDs changed after LoWS. IEDs have been shown to co-occur with the transition from up- to down-state of sleep SWs[26]. In our data, we did not find significant changes in IED rates following LoWS (Fig. 3d, e) or pSW (Supplementary Fig. 8a, b), in contrast to sleep SW[26].

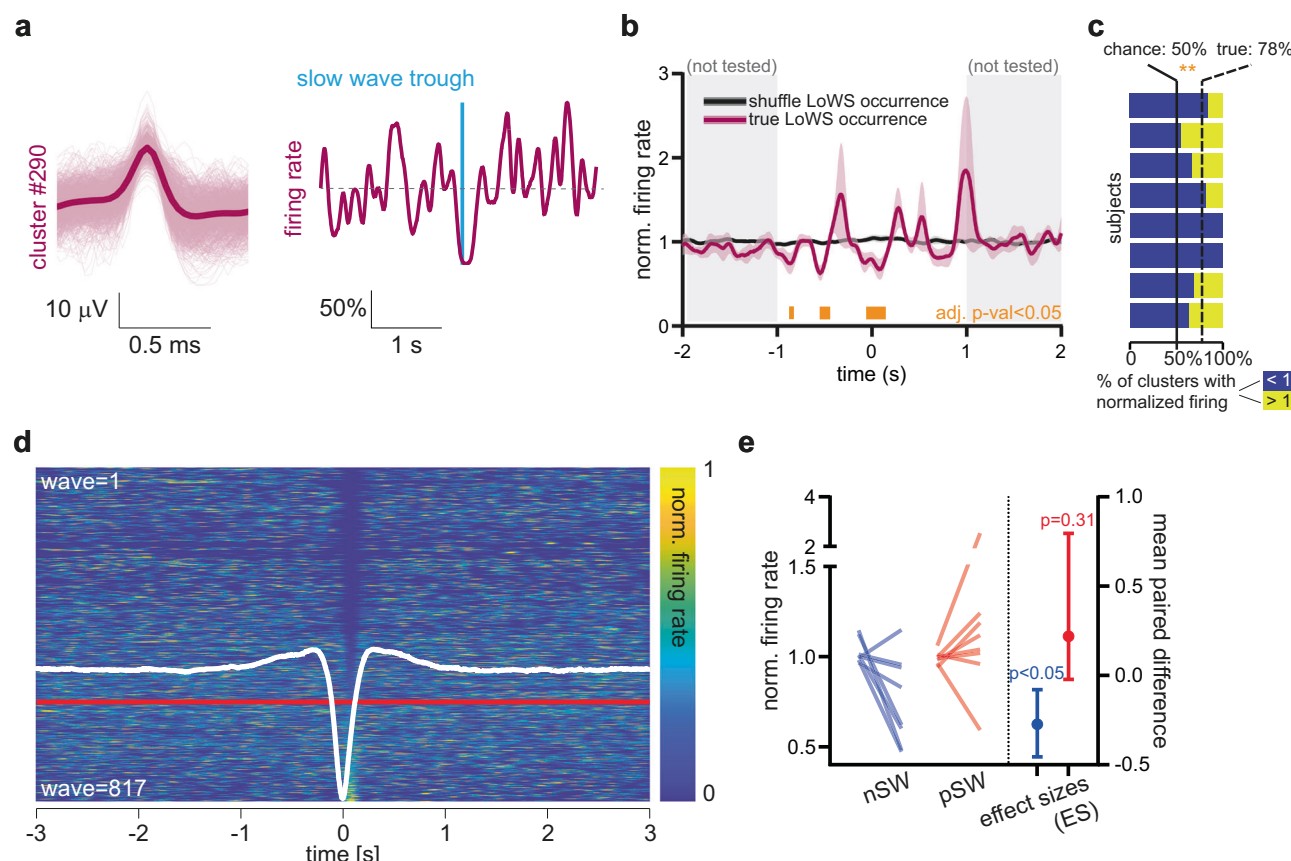

**Fig. 2 | Local wake slow wave (LoWS) trough is associated with a down-state of neuronal activity. a** *Left* Example of 1 unit. *Right* Normalized mean firing rate of that unit aligned to all LoWS trough (blue line). There is a decrease in firing locked to LoWS trough. Scale: percentage of the mean. **b** Mean firing rate, normalized by the mean firing rate per epoch, around the trough of LoWS (mean ± SEM, $n = 8$). Pink: actual neuronal firing ±SEM; black: neuronal firing around shuffled LoWS occurrence ±SEM. Orange boxes: periods with significant differences. **c** The percentage of units, by patient, that shows a decrease in firing rate (dark blue), at the minimum identified in **b** is significantly above 50% (two-sided, one-sample $t$-test, **p = 0.0024, $n = 8$ patients). **d** Neuronal firing rate around LoWS (see Online Methods). The firing rate is normalized to the mean firing rate per epoch (the

maximum being set to 1 for display) and then sorted by the firing rate at the minimum identified in **b**, i.e., 0.079–0.081 s. Below the red line, LoWS with associated firing rate showing an increase in comparison to baseline. White trace: grand average LoWS from Fig. 1b. **e** Paired statistic comparing the mean firing rate locked to randomized nSW occurrence (left end of the blue lines) and to randomized pSW occurrence (left end of the red lines) and mean firing rate locked to nSW and pSW occurrence (right end of the blue and red lines respectively). The period of interest corresponds to the period in-between the two 0-crossings of nSW (−0.15 to 0.165 s). Blue: locking to nSW; red: locking to pSW. There is a significant decrease around nSW (two-sided permutation paired $t$-test, $p = 0.0368$, $n = 8$ patients, ES: mean and 95% CI).

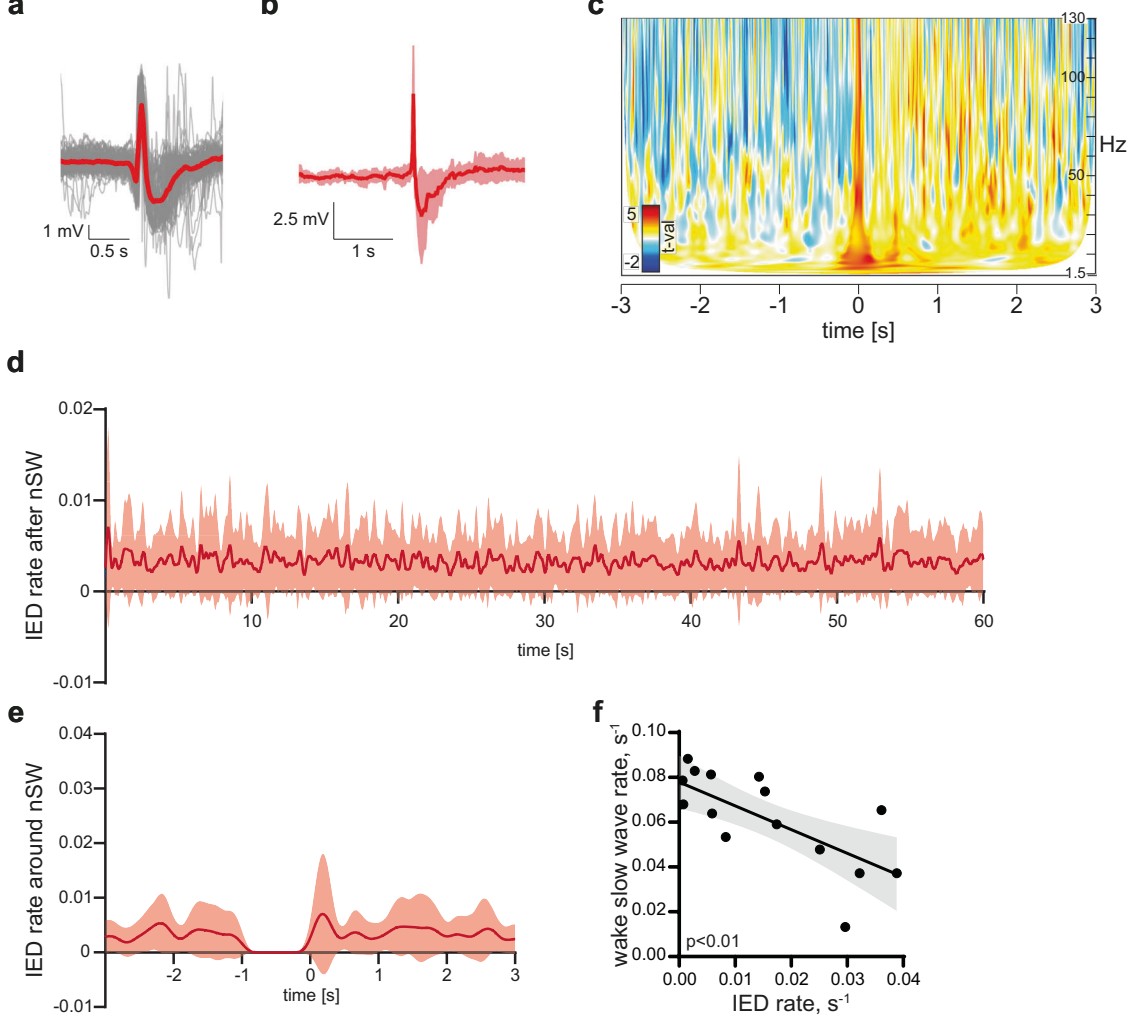

**Fig. 3 | Interictal epileptiform discharges (IEDs) are not activated by LoWS.**
**a** Identified interictal epileptiform discharges (IEDs) in one patient (patient 1, electrode 2, 94 IEDs, mean in red). **b** Grand average (±SD) IED across patients. **c** Time-frequency decomposition of IEDs converted into *t*-statistic values. The presence of LoWS is not associated with a long-term (**d**) or short-term (**e**) change in the incidence rate of IEDs (per s, per electrode and per nSW). Both panels show mean ± SD. See also Supplementary Fig. 8. **f** Rate of LoWS is inversely correlated with the rate of IEDs (Pearson correlation, two-sided *p* = 0.002, best linear fit and 95% confidence bands).

Reciprocally, we did not observe a change of LoWS rate before IEDs (Supplementary Fig. 8c, d). We then compared the overall rate of LoWS with that of IEDs. We observed a significant negative correlation across patients (Pearson correlation, $r^2 = 0.51$, $p = 0.002$, n = 16, Fig. 3f, see Discussion for further comments). We hypothesized that this might explain why we found more LoWS in our dataset than in the Boran dataset. Indeed, after correcting the rate of LoWS by that of IEDs (see Online Methods), we found a similar normalized rate of LoWS in both datasets (Boran versus UCL, ES, 95% CI: −0.0004, −0.001 to 0.0001, n = 9 and n = 16 respectively, p = 0.22, Supplementary Fig. 1b, two last columns). Besides incidence rate, neither the slope (Pearson correlation, $r^2 < 0.001$, $p > 0.99$, $n = 16$, Supplementary Fig. 9a), nor the amplitude (Pearson correlation, $r^2 < 0.001$, $p = 0.96$, $n = 16$, Supplementary Fig. 9b) of LoWS correlated with the IED rate.

Some evidence indicates that IEDs are preceded by a progressive increase in network excitability[34,35]. In line with this, we observed a progressive increase in HG power prior to IEDs (β-estimate ± sem: increase of $1 \log(\mu V^2)$ in HG power each $66.5 \pm 25.4$ s, $n = 15$ patients, $p = 0.009$, Fig. 4). Thus, in parallel to this progressive increase in excitability before IEDs, we asked whether LoWS undergo electro-physiological changes similar to those of sleep SW in response to

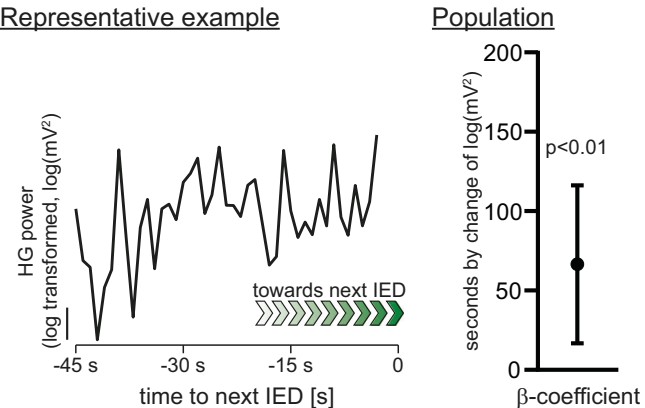

**Fig. 4 | Increases in high-gamma (HG) power anticipate interictal epileptiform discharges (IEDs).** Left: representative example showing that baseline high-gamma (HG) power increases with temporal proximity to the next interictal epileptiform discharge (IED). Right: β-estimate (standardized regression coefficient) and 95% CI across participants, confirming that there is a significant progressive increase in baseline HG prior to IEDs (linear mixed model, *p* = 0.009, *n* = 15 patients).

increased excitability after a period of wakefulness. Under high homeostatic sleep pressure, sleep SW present with high slope and amplitude[15] (see Fig. 1a, b). Using these features as independent variables in a linear mixed model with time to the next IED as the dependent variable, we found that the slope (β-estimate ± sem: slope increases by 0.001% ± 0.0002% each 1 s, adj. $p < 0.0001$, $n = 16$, Fig. 5a, b) and amplitude (β-estimate ± sem: amplitude increases by 0.002% ± 0.0001% each 1 s, adj. $p < 0.0001$, $n = 16$, Fig. 5a, c) of LoWS increased significantly across time leading up to the next IED. Importantly, although these rates of change may appear small, they are within the range of that reported in sleep SW across a whole night (at least for SW slope: change of 20–30% across 381 min of sleep)[13]. Thus, two classical markers of synaptic strength (the slope and amplitude of LoWS)[6] change in parallel to progressive increases in HG power prior to IEDs. This indicates that LoWS exhibit the first property expected of a homeostatic process: they show adaptation to increased network excitability, similar to the response of sleep SW to increased activity during wakefulness.

## Effects of LoWS on IEDs

The results above indicate a response of LoWS to increases in network excitability leading up to IEDs, but do LoWS also represent a mechanism to reduce these aberrant increases in network activity? To investigate this question, we analyzed the changes in HG power during IEDs, as a surrogate for excitability during IED, and related these to the delay since the last LoWS. We used a linear mixed model with delay since the last LoWS as the independent variable and HG power during IEDs (from

−0.05 to 0.05 s around peak amplitude) as the dependent variable. We found a significant effect of the delay since the last LoWS on HG power during IEDs (β-estimate ± sem: $2 \cdot 10^{-3} \pm 3 \cdot 10^{-4}$ log($\mu V^2$) s$^{-1}$, $n = 16$, $p < 0.0001$, Fig. 5d), indicating that for every additional 1 s since the last LoWS, HG power during IEDs increases by $-1 \mu V^2$, or 0.9% of the grand average HG power across patients (mean ± SD: $113.1 \pm 56.7 \mu V^2$, $n = 17$ patients). This suggests that any beneficial impact of LoWS on IEDs operates as a dynamic process and dissipates with time. In sum, these results indicate that LoWS also exhibit the second property expected of a homeostatic process: they reduce network excitability, similar to the proposed function of sleep SW.

## Phasic modulation of LoWS during a cognitive task

Cognitive tasks are known to increase the brain's metabolic demand[36]. We therefore asked whether this increased demand also changed features of LoWS, consistent with a putative homeostatic role. To do so, we examined the properties of LoWS during the encoding and retrieval phases of a temporal lobe dependent associative memory task (see Online Methods). We found that the rate of LoWS increased during encoding (ES, 95% CI: 0.0116, 0.0038–0.026 s$^{-1}$ electrode$^{-1}$, $p = 0.023$, $n = 17$, Fig. 6a) but not retrieval (ES, 95% CI: 0.00446, −0.00343 to 0.0136 s$^{-1}$ electrode$^{-1}$, $p = 0.348$, $n = 17$, Fig. 6a). Furthermore, we observed that the slope of LoWS was steeper during both the encoding (ES, 95% CI: 0.056, 0.03–0.0936 a.u., $p = 0.002$, $n = 17$, Fig. 6b) and retrieval (ES, 95% CI: 0.0504, 0.0178–0.0805 a.u., $p = 0.0074$, $n = 17$, Fig. 6b) phases of the memory task, while LoWS amplitude was higher during encoding (ES, 95% CI: 0.108,

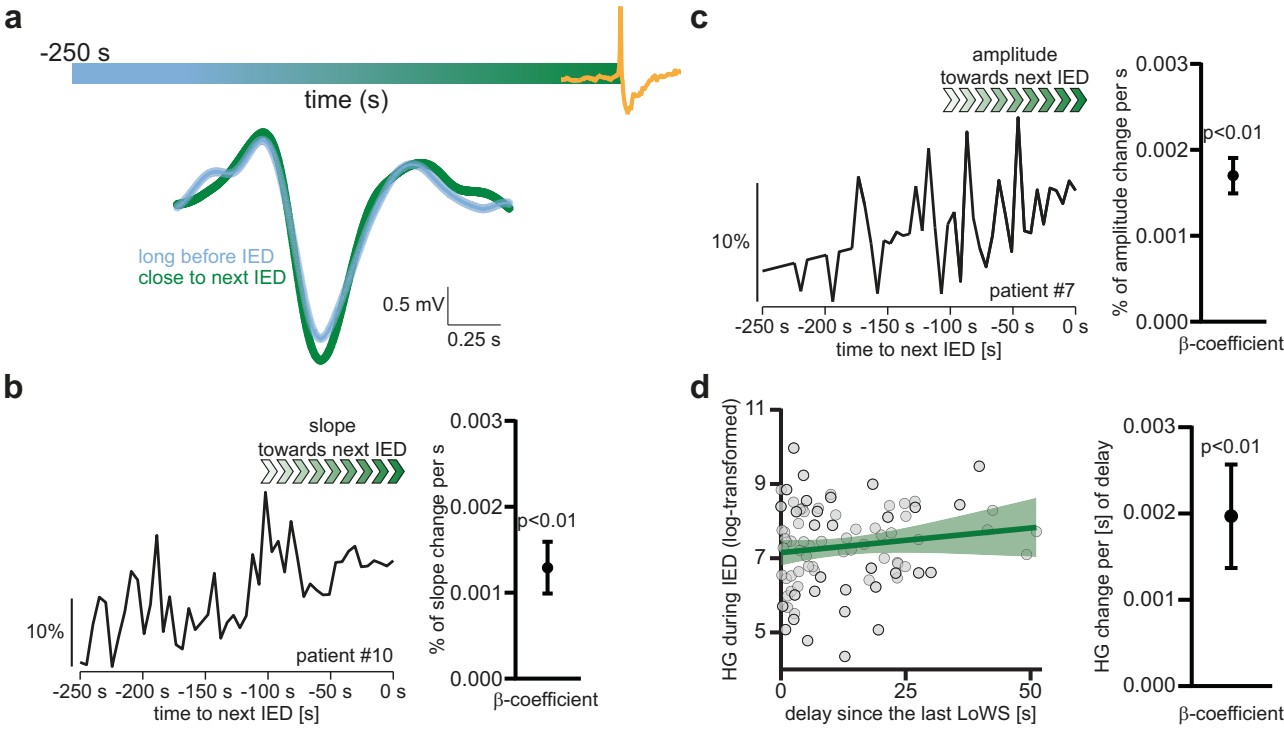

**Fig. 5 | LoWS adapt their slope and amplitude prior to interictal epileptiform discharges (IEDs), and their timing relative to the next IED impacts on the excitability of IEDs. a** *Left* Grand average LoWS 250 s before interictal epileptiform discharges (IEDs) (light blue) and 5 s before IEDs (green), illustrating that the closer to the next IED, the larger the slope and amplitude of LoWS. The color bar above the traces represents the time-axis, with the IED in yellow. Representative examples of the progressive increase in the slope (**b**) and amplitude (**c**) of LoWS before the next IED. Scale: % of the maximal absolute slope and amplitude respectively. Next to each trace, the bar plots show the β-estimate and 95% confidence interval of the linear mixed model across patients. Scale: % of change of slope and amplitude per s. This confirms a significant increase of the slope

(linear mixed model, Bonferroni−Holm corrected $p < 0.0001$, $n = 16$ patients) and amplitude (linear mixed model, Bonferroni−Holm corrected $p < 0.0001$, $n = 16$ patients) of LoWS before the next IED. For display only, missing datapoints have been added using linear interpolation. **d** Left: representative example of the relationship between delay since the last LoWS and high-gamma (HG) power during the index IED: the longer the delay since the last LoWS, the higher the HG power during IEDs (best linear fit and 95% confidence bands). Right: significant regression across participants between delay since the last LoWS and HG power during the index IED (linear mixed model, β-estimate and 95% CI, $p < 0.0001$, $n = 16$ patients). Scale: change in HG power (log-transformed) per unit of time (in s) since the last LoWS.

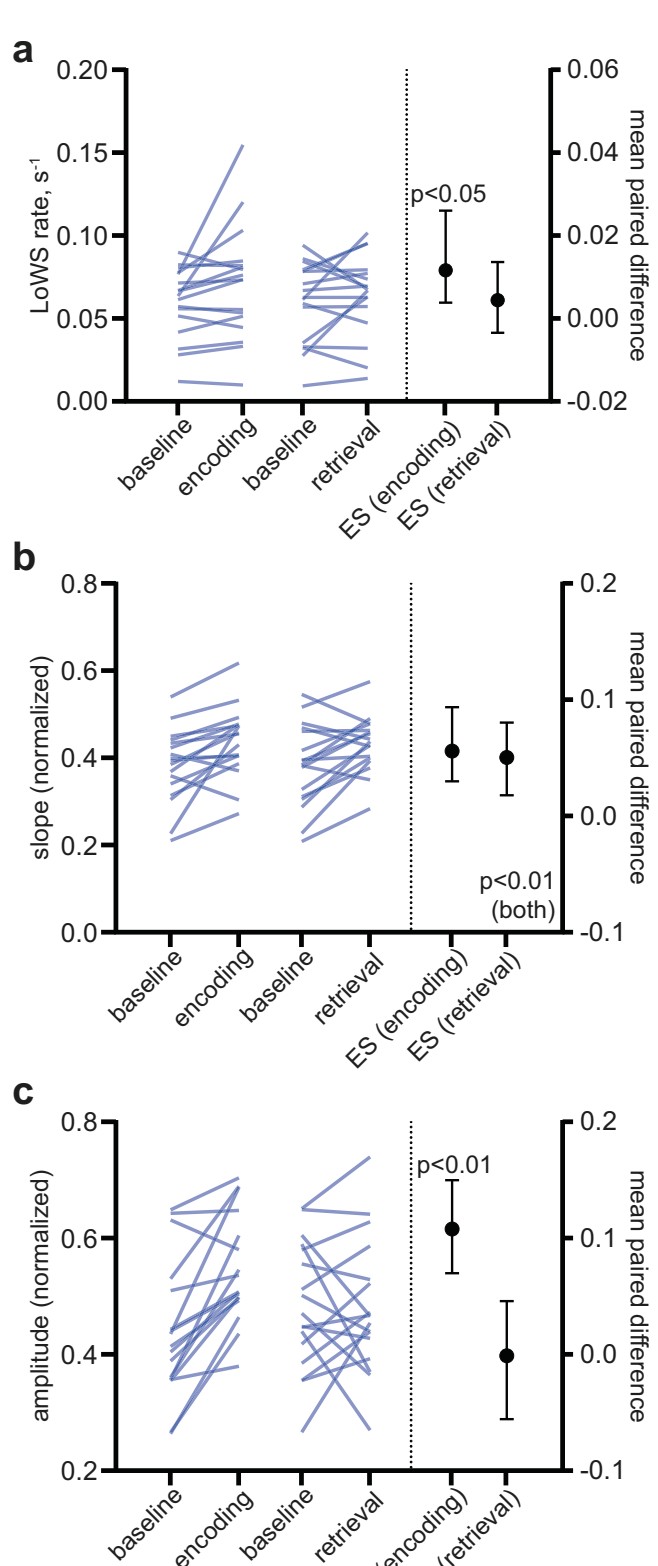

**Fig. 6 | LoWS adapt their rate, slope and amplitude during the cognitive task.**
**a** The rate of LoWS increases during the encoding phase of the task (two-sided permutation paired *t*-test, *p* = 0.023, *n* = 17 patients, ES: mean and 95% CI). **b** The slope of LoWS increased during the encoding (two-sided permutation paired *t*-test, *p* = 0.002, *n* = 17 patients, ES: mean and 95% CI) and retrieval phases (two-sided permutation paired t-test, *p* = 0.007, *n* = 17 patients, ES: mean and 95% CI) of the task. **c** The amplitude of LoWS increased during the encoding phase of the task (two-sided permutation paired *t*-test, *p* = 0.0002, *n* = 17 patients, ES: mean and 95% CI). ES effect size.

0.07−0.15 a.u., *p* = 0.0002, *n* = 17, Fig. 6c) but not retrieval (ES, 95% CI: −0.00104, −0.0555 to 0.0459 a.u., *p* = 0.972, *n* = 17, Fig. 6c) phase. Together, these results support an adaptative modulation of LoWS by cognitive processes.

### LoWS are associated with transient cognitive impairment
One reason why SW are believed to be restricted to sleep is their impact on neuronal activity, which might be expected to have a detrimental effect on cognition[5]. We tested this prediction by analyzing the relationship between LoWS and performance on the associative memory task. Using a linear mixed model, we found that a higher rate of LoWS during retrieval, but not encoding, was associated with longer reaction times (RTs; β-estimate ± sem during retrieval: 0.56 ± 0.14 s², *n* = 17, *p* < 0.0001, Fig. 7a; during encoding: −0.33 s² ± 0.31, *n* = 17, *p* = 0.84). This indicates that the RT was prolonged by 0.56 s for each increase of 1 LoWS per s. To address the possibility that this relationship arose from transient periods of drowsiness or fluctuations in attention (which could increase both RTs and the incidence of LoWS), we repeated the analysis using accurate trials only (i.e., correct answers) and again found a significant correlation (β-estimate ± sem: 0.51 ± 0.17 s², *n* = 17, *p* = 0.002). Conversely, the rate of LoWS during encoding and retrieval did not differ between high and low accuracy trials (ES, 95% CI: encoding: −0.004, −0.07 to 0.06 s⁻¹, *n* = 15 patients, *p* = 0.919; retrieval: 0.004, −0.04 to 0.04 s⁻¹, *n* = 17, *p* = 0.841), indicating that these events were associated with impaired processing speed rather than accuracy. Performing the same analysis on IEDs revealed no significant association between incidence rate and RTs (β-estimate ± sem during encoding: −0.01 ± 0.98 s², *n* = 17, *p* = 1; during retrieval: 0.70 ± 0.48 s², *n* = 17, *p* = 0.14). However, the rate of IEDs was significantly lower during high versus low accuracy trials in both the encoding and retrieval phases (ES, 95% CI during encoding: −0.0413, −0.122 to −0.0132 s⁻¹, *p* = 0.0114, *n* = 15 patients; ES, 95% CI during retrieval: −0.0187, −0.0319 to −0.00805 s⁻¹, *p* = 0.0032, *n* = 16, Fig. 7b), consistent with previous studies[37]. Altogether, these results indicate that LoWS and IEDs are associated with distinct cognitive impairments: the former with increased reaction time and the latter with decreased accuracy.

## Discussion
In sum, these findings reveal the existence of SW during wakefulness that recapitulate the distinctive features of sleep SW, including a down-state of neuronal activity. We demonstrate that these SWs are distinct from and not the result of sporadic increases in delta power that can occur as a result of lesions or within epileptogenic foci. Although evidence supports a restorative function of SW during sleep[10,11], it was previously unknown whether such beneficial activity could occur during wakefulness under pathological conditions. Here, we provide evidence of changes in LoWS properties (slope and amplitude) before IEDs that reproduce the changes of sleep SW under high homeostatic sleep pressure[13,15]. We further show that the closer an IED is to the preceding LoWS, the lower the associated network excitability (measured by HG power). We therefore propose that LoWS operate as key components of epilepsy homeostasis[38], mirroring the well-known sleep homeostasis regulated by sleep SWs[6]. This is further supported by the negative correlation between LoWS rate and IED rate, which suggests that IEDs occur more frequently in patients with fewer protective LoWS. Lastly, we observed that these beneficial effects are associated with a negative effect on cognitive processing, with slower RTs in an associative memory task, consistent with the impact of SW on neuronal activity[5].

While it is difficult to infer whether LoWS are physiological or induced by the presence of epilepsy, we do not expect them to be entirely specific to epilepsy. Indeed, we were able to identify LoWS in presumably healthy brain regions within the MNI dataset; and furthermore, they do not show evidence of co-localization with the

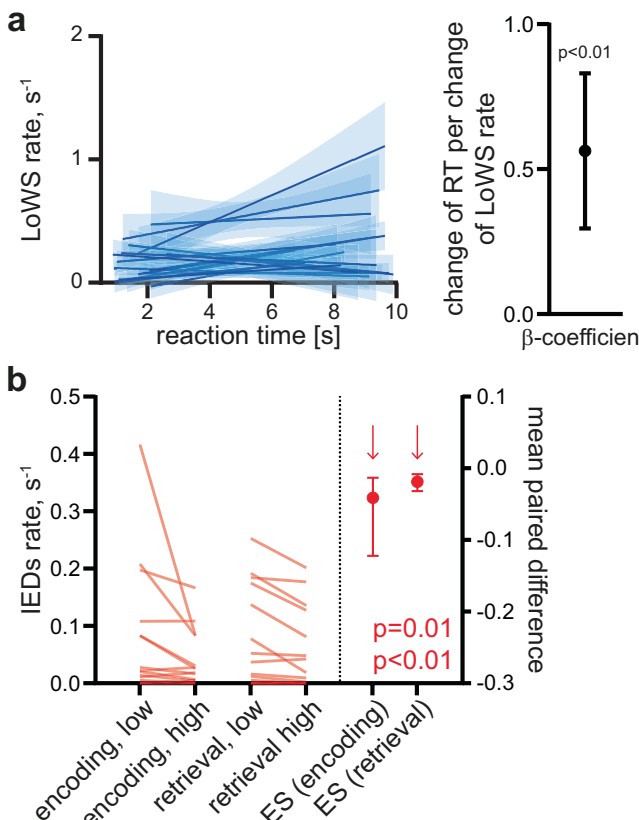

**Fig. 7 | Local Wake Slow wave (LoWS)- and interictal epileptiform discharge (IED)-associated transient cognitive impairment. a** *Left.* Local Wake Slow wave (LoWS) rate during retrieval predicts reaction time (RT). *Right.* Across patients, the β-estimate and 95% CI of the linear mixed model between RT (dependent variable) and LoWS rate (independent variable) is significant (linear mixed model, $p < 0.0001$, $n = 17$ patients). Scale: change of RT (in s) per unit increase of LoWS rate (in $s^{-1}$). **b** Interictal epileptiform discharge (IED) rate is lower during encoding and retrieval periods of trials with high accuracy (two-sided permutation paired $t$-test during encoding: $p = 0.0114$, $n = 15$ patients, ES: mean and 95% CI; retrieval: $p = 0.0032$, $n = 16$ patients, ES: mean and 95% CI). ES effect size.

epileptic focus. Hence, our interpretation is that they represent, at least in part, physiological activities that impact on neuronal excitability, including that associated with epileptiform discharges.

What could be the underlying mechanism for a protective effect of LoWS? Evidence suggests that SW are instrumental in decreasing neuronal excitability during sleep[4,12]. In particular, sleep SW orchestrate increased neuronal synchronization (i.e., low frequency bursts of population activity) that has been shown to promote synaptic downscaling (i.e., a decrease in synaptic strength)[8]. Since neuronal synchronization has been shown to correlate with sleep SW slope[12], the increase in LoWS slope that we observed before IEDs (although relatively small across this much shorter timescale) might also reflect increased neuronal synchronization, which promotes synaptic downscaling and could therefore underlie the reduction in HG power during subsequent IEDs. Further studies, including manipulation protocols, will be necessary to demonstrate causality.

SWs have traditionally been considered to be specific to, and almost exclusively studied during sleep[4,6,10,12,15,18,29,39,40], whilst evidence of slow wave activity during wakefulness has only recently been described[25,41,42]. Those studies of wake SWs have made use of sleep-deprived animals[43], humans under sleep pressure undergoing scalp EEG recordings[41], stimulation of cortical regions surrounding focal brain injury[24], or thermocoagulation of clinically-defined brain areas[25]. However, typical slow waves with an associated neuronal down-state

have not previously been described in awake humans, and it was unknown whether this core feature of sleep SW occurred during wakefulness. Indeed, the identification of neuronal down-states has been recognized as particularly challenging in awake humans, since these periods are expected to be very short[44]. Hence, LoWS might previously have been overlooked, but could be particularly important in the context of brain pathology[24,25,43]. Nonetheless, our results are based on wake SW with a particularly high amplitude, detected by an algorithm similar to that used by Frauscher et al.[26], in order to distinguish SW from background noise. It thus remains unclear if the effects we observe generalize to all (including lower amplitude) SWs.

Although different frequency ranges have been used to define sleep SW, we followed the range typically used for SW detection (0.5–4 Hz)[13,26,28] and our detection method was validated by the presence of an associated down-state. To note, we excluded post-IED slow waves, since evidence suggests that sleep slow waves participate in sleep homeostatic regulation[17] and are different from post-IED waves[28]. Nevertheless, further work is needed to fully understand the properties that differentiate post-IED waves from LoWS.

Previous intracranial EEG studies in people with epilepsy describe separate low and high theta bands that are independently modulated by various cognitive processes[45–47]. However, the results presented here raise the possibility that changes in low theta power may reflect an increased incidence of LoWS associated with cognitive effort. This would also explain why low theta is rarely observed in EEG or MEG recordings from healthy participants performing similar tasks[48] who may be less likely to have LoWS.

The LoWS we observed in epilepsy resulted in decreased HG activity and neuronal firing akin to an interruption of network function, so it is not surprising that LoWS were also associated with slower memory retrieval. Using scalp EEG, a previous study suggested that when patients declare thinking about nothing, or something other than the current cognitive task, they present an increased density of SWs[41]. However, it remains unknown whether this subjective experience reflects the occurrence of SWs, or whether reduced attention triggers the expression of SWs. In our study, the finding that LoWS rate predicts RTs, even when analyzing correct trials only, suggests that this relationship is independent from fluctuating degrees of attention. Since we used an associative memory task, we focused our analyses on temporal lobe recordings. However, the impact of LoWS on other brain regions might vary, depending on the degree of the associated modulation of neuronal activity[28,49,50], and should be tested further using different cognitive tasks. Regarding IEDs, their impact on memory when generated in extrahippocampal brain regions has already been addressed[51] and we did not intend to replicate this result in the present study. It would have required more patients to control for the multiple brain regions potentially involved in modulating memory performance and was beyond the scope of this work.

Both beneficial and detrimental or even pro-epileptic functions of SW have been suggested[24–26,43,52,53]. This may, however, be dependent upon brain state and timescale and a clear causal association has not been established. Our findings indicate that the wake slow waves we identified are associated with a reduction in excitability and a beneficial impact on IEDs; and this opens promising therapeutic perspectives. In this context, the specificity of low-frequency (1–3 Hz) stimulation to elicit synaptic depression in vitro[54] and to decrease cortical excitability in humans[55] as well as its antiepileptogenic effect in a rodent model of epilepsy[56] fits well with the hypothetical anti-epileptic property of LoWS. Our work proposes that the same pattern of activity is naturally employed by the brain in patients with epilepsy, and a failure to express LoWS could be detrimental. Modulating LoWS could thus represent an efficient neurostimulation protocol that enhances their protective activity to increase the epileptic threshold. Preliminary work has shown that it is possible to experimentally

amplify sleep SW[57], and the application or improvement[58] of such devices to the field of epilepsy could be within reach in the near future.

To summarize, our study provides evidence of "micro-sleep" modules during wakefulness in epilepsy that present with clear homeostatic features, showing not only adaptation in association with increases in network excitability before IEDs, but also negative feedback on IEDs, at the cost of cognitive lapses. Our work highlights the compromise made by a brain that is affected by pathological activity and calls for the development of interventional strategies aimed at promoting LoWS to test their direct therapeutic potential in epilepsy.

## Methods
### Patients
A total of 25 patients with medically refractory epilepsy (11 female, 23 right-handed, mean age of 38.5 yrs, 17 with macroelectrode recordings and 8 with microelectrode recordings, Supplementary Table 1) undergoing intracranial EEG monitoring for clinical purposes at the National Hospital for Neurology and Neurosurgery, London, were included in this study. One supplementary patient with microelectrode recording was excluded because of high epileptic activity that would have impeded a reliable analysis of slow wave activity. Patients with microelectrode recordings also had macroelectrodes implanted, but there were technical issues with the synchronization of both signals which prevented us from including those macroelectrode data. Prior approval was granted by the NHS Research Ethics Committee (15/LO/1783), and informed written consent was obtained from each patient. Patients were recruited if the research protocol did not impact negatively on their clinical care, and if intracranial electrodes were targeted at mesial temporal lobe structures. Sex and gender were not considered in the study design, given the rarity of the data. Some of these patients participated in other research projects that have been published elsewhere[59,60].

### Task
A description of the associative memory task is provided in more detail elsewhere[59,60]. Briefly, the task was divided into encoding and retrieval phases. During encoding, participants were presented with 27 pairs of images that remained on screen for 6 s, preceded by a 2 s fixation cross and followed by a 2 s blank screen. Unbeknownst to participants, each pair was drawn from one of 9 discrete events comprised of a person, place and object. As such, each item (person, place, object) was paired with two other items over the course of encoding. During retrieval, participants were first presented with a single cue image for 3 s, drawn either from one of the 9 events viewed during encoding or an equal number of New events, comprised of person, place and object images that had not been viewed during encoding. Next, participants were asked whether the cue image was Old or New? – i.e., whether that image had formed part of a pair viewed during encoding or not. If the cue image was Old, whether or not the participant correctly identified it as such, they were then successively asked to indicate which two images it had been paired with, from two groups of four alternative choices drawn from the same image category across all Old events (i.e. if cued with a person, they were first asked which place and then which object it had been paired with during encoding). In each case, participants had unlimited time to indicate their response with a key press, and reaction times (RTs) were recorded. Importantly, the rate of LoWS/IEDs during retrieval was correlated with the accuracy of memory retrieval in that trial, whilst the rate of LoWS/IEDs during encoding was correlated with the accuracy of memory retrieval across both retrieval trials for the pair being presented on screen. This leads to a small difference in the number of patients included in each analysis.

### Data acquisition
Macroelectrode EEG data were recorded continuously against a common reference located in white matter remote from the epileptic focus at a sample rate of 2048 Hz (Patients 12 and 13), 512 Hz (Patients 1 and 16), or 1024 Hz (all other patients), with a low-pass hardware filter set at a cut-off frequency equal to ¼ of the sampling rate and a 0.15 Hz high-pass, using a Micromed SD long-term monitoring system (Micromed). Electrodes (platinum Spencer probes) were implanted purely for clinical purposes via stereotaxis. Recordings made at a higher sampling rate were downsampled to 512 Hz before any analyses were performed. Microwire EEG data were recorded continuously at a sample rate of 30 kHz using a Blackrock Neuroport system (Blackrock Neurotech LLC) and Adtech Behnke Fried electrodes.

Electrode locations were established visually by a consensus between authors (LS, RR and DB) combining the pre-implantation MRI and post-implantation CT using EpiNav™. Further information can be found in Supplementary Table 1. For the purpose of this study, we only analysed data from electrodes located in the hippocampus, amygdala and temporal neocortex, so that we could focus on the interaction between IEDs, LoWS, and associative memory function.

Patients were alert and responsive during cognitive task periods, as assessed directly by one of the co-authors (UV, JB, or DB) in the recording room. Between cognitive task periods, patients were seen intermittently, and no evidence of sleep was noted. Furthermore, since some recordings were particularly long (see Supplementary Table 1), we confirmed that there were no signs of drowsiness in the EEG by comparing delta (0.5–4 Hz), theta (4.1–7 Hz) and alpha (7.1–12 Hz) power during vs outside the cognitive task periods (Supplementary Fig. 10a). It is possible that the trend for an increase in delta power during the cognitive task reflects the increased incidence of LoWS (although, as mentioned earlier, there was no correlation between delta power and LoWS rate, Supplementary Fig. 3a), or could be a consequence of memory processing[45]. A standard criterion of NREM 1 is a decrease in alpha peak frequency by 1 Hz[61]; but in our data, alpha peak frequency was actually significantly higher during rest periods than during the cognitive task (during rest, mean ± SD: 9.74 ± 0.6 Hz; during cognitive task: 9.61 ± 0.58 Hz, paired $t$-test, $p = 0.0262$, Supplementary Fig. 10b). Hence, the rest period was not marked by two fundamental properties of NREM 1 (a decrease in alpha power and peak frequency), and delta/theta activity also tended to be higher during rest periods, implying that patients did not sleep between cognitive task periods.

### Publicly available datasets
Two publicly available datasets[30–33] relating to complementary cohorts from other centers (total of 115 supplementary patients), were also included for control purposes. The Boran dataset[30] was published under a CC BY-SA 4.0 license (https://creativecommons.org/licenses/by/4.0/) and is available here: https://doi.org/10.12751/g-node.d76994. The MNI dataset[31–33] is publicly available at https://mni-open-ieegatlas.research.mcgill.ca/. In the Boran dataset, we analyzed hippocampal activity in intracranial recording from patients with epilepsy undergoing a working memory task[62]. Data were acquired at 4 kHz (passband 0.5–1000 Hz, Neuralynx ATLAS) against a common intracranial reference and resampled to 2 kHz[30]. Further information can be found in[30]. In the MNI Open iEEG dataset, we analyzed activity from all contacts in intracranial recordings from patients across the sleep-wake cycle (wakefulness, NREM 2, NREM 3 and REM sleep)[31–33]. Data were resampled to a sample rate of 200 Hz (low-pass filter set at 80 Hz). These data are provided in a bipolar montage (see note regarding the choice of the reference in the subsection "Identification of slow waves" below). Further information can be found in refs. [31–33].

### Code and software used for analyses
Analyses were performed using custom Matlab (The MathWorks, v. 2022a) code and the Fieldtrip toolbox[63]. Statistical analyses were performed using the web application EstimationStats[64], Graphpad Prism

(v. 9.5.0), SPSS (v. 29.0.0.0) and Matlab (v. 2022a). Figures were designed in Matlab, Graphpad Prism and Adobe Illustrator.

## Identification of IEDs

IEDs were identified in a semi-automated manner. First, the data were screened using a published algorithm for rodent models of epilepsy[65] that was adapted to human recordings. Briefly, data were filtered between 20–80 Hz (using a second order Butterworth filter) and the amplitude of the filtered signal estimated using the absolute value of the Hilbert transform. Candidate IEDs were then defined as periods where both the amplitude of the filtered signal exceeded 3.5 times the mean envelope and the amplitude of the unfiltered signal exceeded 4 times the mean envelope. Next, if >1 recording contact in any single region of interest (hippocampus, amygdala, temporal neocortex) exhibited a candidate IED within a window of ±0.3 s, only the candidate IED with the highest amplitude was retained. This was introduced to avoid working on replicates and/or volume conducted activity. Finally, all candidate IEDs were curated by visual inspection and missed IEDs were manually added.

It is important to note that we generally present the average rate of IEDs per electrode, which can appear low. This is to avoid patients with more intra-hippocampal electrodes presenting with an artificially higher rate of IEDs. Uncorrected, the maximal rate reaches $0.23\,\mathrm{s}^{-1}$ (mean ± SD: $0.05 \pm 0.07\,\mathrm{s}^{-1}$).

## Identification of slow waves

Slow waves were detected using a standard algorithm[13,26,28,29]. First, the LFP signal was filtered between 0.5–4 Hz (using a second order Butterworth filter), all positive-to-negative zero-crossings and negative-to-positive zero-crossings in the filtered signal were detected, and events with a duration of 0.25 to 1 s were selected as candidate slow waves. Next, any candidate slow waves that occurred ≤1 s after a detected IED on any of the recording contacts in the same brain region were discarded. Finally, only the remaining candidate slow waves whose amplitude exceeded the 90th percentile were retained. Candidate slow waves were then visually inspected, and any that appeared artifactual or had an apparent IED occurring shortly beforehand were discarded. We were concerned by the fact that IEDs are typically followed by a slow wave, but it was not possible to visually inspect all slow waves, due to the large number detected. In addition to the precautionary measures taken above, we thus undertook several verification steps to ensure that detected slow waves were not mislabeled IEDs. First, the detected slow waves were visually similar to those described in other publications (Fig. 1c, d)[13,29]. Second, detected slow waves and IEDs had a different spectral signature (Figs. 1f, g and 3c). Third, we observed that 45–130 Hz high-gamma (HG) power during IEDs (measured either during the time window used for down-state analysis, −0.15 to 0.165 s; or during the time window of IED peak, −0.05 to 0.05 s) was significantly higher than during pSW (ES, 95% CI: $4.49 \cdot 10^3$, $2.68 \cdot 10^3$ – $7.25 \cdot 10^3\,\mu\mathrm{V}^2$; and ES, 95% CI: $2.24 \cdot 10^4$, $1.29 \cdot 10^4$ – $3.71 \cdot 10^4\,\mu\mathrm{V}^2$, $p = 0.0008$ for both, Supplementary Fig. 7c). Note that since HG power is decreased during nSW, we did not compare it with HG increases during IEDs. Fourth, we reasoned that if slow waves were mislabeled IEDs, then variability in the rate of slow waves across electrodes should follow variability in the rate of IEDs rate across the same electrodes. To test this, we defined an index of parallel rate variability (PRV) that records the ratio of IED rate to slow wave rate for each electrode minus the ratio of the sum of IED rate and slow wave rate across all electrodes for each participant. As such, the median PRV value is expected to be significantly different from 0 if the rates of IEDs and slow waves do not follow the same pattern across electrodes. The PRV is defined by:

$$\mathrm{prv} = \sum_{n=1}^{N} \left( \frac{\mathrm{rate(ied)}_{e_n}}{\mathrm{rate(sw)}_{e_n}} - \frac{\mathrm{rate(ied)}_{e_n} + \mathrm{rate(ied)}_{e_{n+1}} + \mathrm{rate(ied)}_{e_N}}{\mathrm{rate(sw)}_{e_n} + \mathrm{rate(sw)}_{e_{n+1}} + \mathrm{rate(sw)}_{e_N}} \right)^2 \quad (1)$$

Where $n = 1$ to $N$ indexes the electrodes in a specific patient. Across participants, the median PRV was significantly different from 0 (median, 25–75th percentile: 0.06, 0.001–1.62, one sample Wilcoxon test, $p < 0.0001$, Supplementary Fig. 11), indicating that variability in the rate of IEDs rate is not paralleled by variability in the rate of slow waves across electrodes. We calculated the PRV only in the hippocampus, where all main analyses were performed. Fifth, we reasoned that if LoWS were mislabeled IEDs, then electrodes that exhibit the greatest number of IEDs should also record the greatest number of slow waves. However, there was no difference in the rate of slow waves between the electrode with the highest rate of IEDs and all other electrodes (ES, 95% CI: −0.002, −0.01 to $0.003\,\mathrm{s}^{-1}$, $p = 0.595$, Supplementary Fig. 12). Sixth, we observed that LoWS and IEDs are associated with different memory deficits (Fig. 7). Seventh, we observed that LoWS and IEDs rates are negatively correlated (Fig. 3f). Conversely, if LoWS were a subpopulation of mislabeled IEDs, and since LoWS detection is semi-automatic (except for the visual removal of artefacts and post-IED waves), then the proportion of mislabeled IEDs as LoWS should be constant across participants, resulting in a positive correlation (Supplementary Fig. 13). Finally, we also wished to address the possibility that the slow waves we identified corresponded to low frequency post-IED waves that propagate more readily across electrode contacts than the sharp component of the IED. To test this, we measured the absolute amplitude of the raw LFP signal at the peak of each IED (i.e., the amplitude of the sharp component) across all hippocampal electrodes and the maximum absolute amplitude in the 0–0.5 s window after that peak (i.e., the amplitude of the subsequent slow wave) and then quantified the relative reduction in amplitude of each with distance from the focus. Across 7 adjacent recording contacts (the maximum number available on each depth electrode), there was no significant difference in these normalized amplitudes (Supplementary Fig. 14a, b). Since our main focus was on the deepest (i.e., most medial) recording contacts (located in the hippocampus), we also compared the ratio of amplitudes on the 1st electrode (electrode of interest) and 2nd (adjacent electrode contact) and did not find a significant difference (ES, 95% CI: 0.0391, −0.0513 to 0.153, $p = 0.49$, Supplementary Fig. 14c). In sum, this indicates that the amplitude of post-IED slow waves decreases with distance from the focus at the same rate as the sharp component, arguing against the wider propagation of post-IED slow waves compared to the sharp component which could make post-IED waves appear as LoWS (i.e., without the sharp component).

To note, steps 4 and 5 only indicate that the majority of LoWS are not IEDs, but the convergence of all 8 findings strongly supports the hypothesis that LoWS are not mislabelled IEDs.

## Corrected rate of slow waves

Since we observed a negative correlation between the rate of LoWS and IEDs (Fig. 3f), we computed the rate of LoWS corrected for the rate of IEDs to verify whether variability in the rate of IEDs could explain discrepancies between our dataset and the Boran dataset (Supplementary Figs. 1b and 2 first columns). The corrected rate was calculated by multiplying the LoWS rate by the IED rate (Supplementary Figs. 1b and 2 last columns), to account for the negative correlation between these variables.

## Identification of slow waves in the Boran dataset

We used another dataset of patients with epilepsy recorded with intracortical electrodes[30] to assess the reproducibility of slow waves detection. To identify hippocampal slow waves, we used the Juelich Histological Atlas, which computes a probability that each contact is located in a subset of brain regions. We considered a contact to locate in the hippocampus if the hippocampus was the most probable region and the probability was ≥25%. We used the Matlab mni2atlas.m function for that purpose (https://github.com/dmascali/mni2atlas/releases/tag/1.1).

### Identification of slow waves in the MNI Open iEEG Atlas

We followed the same procedure as described above. For this dataset, however, we did not manually curate identified SWs to remove potential post-IED waves or artefacts for three reasons: (1) this dataset presents only traces from presumably healthy brain regions, including the lack of IEDs[31], (2) we did not want to introduce a bias in the removal of waves across states of vigilance (it is not possible to be blinded to the vigilance state, since many more waves were detected during NREM 3 than wakefulness) and (3) it was simply not conceivable to visually inspect all identified waves in this very large dataset. Besides wakefulness, the MNI Open iEEG Atlas also comprises data during NREM 2, NREM 3 and REM sleep. Since the data are presented in a bipolar montage, we included waves with both negative and positive polarities in these analyses.

### Choice of the reference

Signals of the Boran dataset and our dataset were calculated against a common intracranial reference. Since slow waves reflect ongoing membrane fluctuations[66] and not a spatially-directed current, the use of a common reference allowed inferring whether the polarity reflects hyper- or depolarization[28,67]. This means that for the MNI Open iEEG Atlas, in which signals were in a bipolar montage, we cannot determine whether the waves represent hyper- or depolarization. For this reason, we combined nSW and pSW in this dataset. Since the incidence rate of nSW and pSW was shown to be highly correlated (Supplementary Fig. 5b), we assumed that combining both activities would only globally increase the rate, without bias for a state of vigilance.

### Correlation between LoWS rate and delta/theta power

Delta (0.5–4 Hz) and theta (4.1–7 Hz) power were estimated using Welch's method and then log-transformed. The rate of LoWS was estimated as the number of LoWS per second and per electrode, after correcting for the refractory period of 1 s following identified IEDs (Supplementary Fig. 3a, b).

### LoWS-slope and amplitude before IEDs

To look for changes in the slope and amplitude of LoWS leading up to IEDs, we first identified the minimum of the 0.5–4 Hz filtered signal in a 1 s window following LoWS onset (which was defined as the amplitude) and defined LoWS slope as the ratio between that amplitude and the post-onset time of this minima. Slope and amplitude values were then normalized by the highest absolute slope and amplitude observed in each patient, so that all values lay in the range of 0 to 1. These values were then used as dependent variables in separate linear mixed models. Time to the next IED was included as a fixed factor. We also included time since the start of recording as a random factor, to account for the possibility that any observed changes in slope and amplitude are simply due to the accumulation of sleep pressure over the course of testing. Finally, we included patient identity as a grouping variable, in this and all other linear mixed model analyses. As in other linear mixed models (Figs. 4, 5b–d and 7a), we used a common slope with variable intercept. For display purposes, panels b and c of Fig. 5 were constructed from the average slope and amplitude respectively by bins of 5 s.

To account for the variability across patients in IEDs and LoWS frequency – and therefore, in the range of delays since the last LoWS and number of observations included in these analyses – we used a linear mixed-model.

### Peri-stimuli time histogram

To generate peri-stimuli time histograms (Figs. 1h, i, 3d, e, Supplementary Fig. 8) of nSW incidence rate around pSW (and vice-versa) and SW around IEDs (and vice-versa), and to avoid bias from under sampling the incidence rates at times distant from the triggering event, we undertook a different analysis (this does not apply to slope, amplitude

and HG power before IEDs [see below], since those are not calculated when LoWS are not present). We first convolved the incidence rate of the modulated activity around the index activity (e.g., modulation of the rate of nSW around IEDs) with a Hanning window of 0.5 s and then calculated the mean occurrence of the modulated activity per electrodes and patients around the index activity. The rate was also normalized by the number of epochs of the index activity, explaining why the apparent basal rate is lower than the actual rate of the modulated activity. To assess statistical significance on a short-term scale, we followed the same procedure described to assess the neuronal down-state of LoWS (see below): we downsampled each patient's mean incidence rate of the modulated activity to 10 Hz and computed a $p$-value for time-windows of 2 s (chosen for homogeneity with the analysis of down-state around LoWS; bootstrap, 5000 loops against shuffled position of index activity). For nSW and pSW, this window extends from −1 to 1, while for IEDs it extends from 0 to 2 s after a SW (since, for IEDs, the rate is expected to be equal to 0 in the second before SW given our detection algorithm; see above). To assess statistical significance on a long-term scale (60 s), we calculated the slope (β-coefficient) of IEDs rate change after LoWS and tested whether the mean slope across participants was significantly different from 0 (one sample Wilcoxon test). In this analysis, the timescale of 60 s could thus have included epochs with other index activity before or after the modulated activity and the use of a shorter window (60 s, instead of 250 s) was intended to mitigate this risk.

### HG power before IEDs

We used the Fieldtrip[63] function ft_freqanalysis.m to calculate HG power (45–130 Hz) during non-overlapping windows of 1 s before IEDs (multitaper frequency transformation) and log-transformed the result. To control for potential outliers, we selected the same number of trials for each time-bin prior to IEDs by identifying for each participant the maximal pre-IED duration (which depended on inter-IEDs delay) that would encompass at least 50% of all IEDs. This explains why $n = 15$ in this analysis. HG power was then used as the independent variable and time to the next IED as the dependent variable in a linear mixed model.

### Interaction between LoWS and HG power during IEDs

HG power during IEDs (−0.05 to 0.05 s around IEDs) was used as the dependent variable in a linear mixed model with delay since the last LoWS as the independent variable. We selected HG power during IEDs, rather than the peak amplitude of IED, because the peak amplitude is sampled at only one timeframe, which would not reliably represent the relatively long duration of IEDs (see Fig. 3a–c), and because the peak amplitude can vary with shifts in ongoing baseline activity. Furthermore, HG activity is typically considered a reliable proxy of neuronal activity[68]. Mean HG power across recordings was measured using the same parameters, randomly sampling the recordings with windows of similar size (500 bootstraps).

### Analyses of down-state

To measure changes in HG power (45–130 Hz) associated with LoWS (Fig. 1e), we used the Fieldtrip[63] function ft_freqanalysis.m (multitaper frequency transformation) for the time window between the two zero-crossings of the grand average LoWS (−0.15 to 0.165 s around the LoWS trough). Power values were baseline corrected using average power at each frequency across the −2.9 to −0.2 s and 0.2 to 2.9 s time windows relative to the LoWS trough. As a control, the same analysis was performed relative to a randomly sampled set of time bins (as opposed to the occurrence of genuine LoWS). Importantly, changes in HG power associated with pSW and IEDs were not baseline corrected, because changes in HG power were observed within the baseline time windows (e.g., during the down-state before pSW [Fig. 1g] and following IEDs [Fig. 3c]). For the time-frequency decompositions in Fig. 1f, g (LoWS)

and Fig. 3c (IEDs), dynamic changes in oscillatory power were estimated using a seven-cycle Morlet wavelet transform (implemented using the Fieldtrip function ft_freqanalysis.m) in ±3 s time windows around IEDs and LoWS across a 1.5–130 Hz frequency range. For control purposes, we used the functions randsample.m or datasample.m to randomly generate twice as many time bins (for LoWS) or the same number of time bins (for IEDs), excluding artefactual periods and time windows of ±3 s around another IED or LoWS (for LoWS) or time windows of ±3 s around another IED (for IEDs).

## Micro-electrode recordings

Single and multi-unit activity was identified using the wave_clus toolbox[69]. Briefly, data were filtered in the 300–3000 Hz range and events whose amplitude was within 4–50 standard deviations of the median noise identified as candidate spikes. Spikes were then clustered automatically based on wavelet features of the waveform before single- and multi-unit activity clusters (denoted together as unit spiking activity or "units" in the manuscript) were manually identified based on the inter-spike interval distribution and average waveform. The firing rate of each unit was then convolved with a 0.2 s Gaussian window and normalized to the mean firing rate across epochs (±3 s around individual LoWS). Finally, units with a mean firing rate <2 Hz across the entire recording session were discarded, following[28]. The statistical analysis in Fig. 2c was performed with units that had a mean firing rate >2 Hz across the entire EEG and with units that had a mean firing rate > 2Hz during the surrounding baseline (−2.9 to −0.2 s and 0.2 to 2.9 s) and both came out as significant after Bonferroni–Holm correction. For display (Fig. 2d) we used units with mean firing >2 Hz during the surrounding baseline. The boundary was set at −2 s and 2 s to encompass the LoWS (the mean duration of LoWS, i.e., in-between two zero crossings, was 0.3571 s, which we rounded to 0.4 s).

To obtain a time-resolved evaluation of the firing rate around LoWS, we first obtained the mean firing rate per patient ($n = 8$) and down-sampled these time series to 10 Hz, thus obtaining a resolution of 0.1 s for statistical analysis. We then used a bootstrapping approach as follows: for each time bin from −1 to 1 s around the LoWS trough, the difference between the mean firing rate around LoWS and five times as many randomly sampled time bins was calculated. Across 5000 bootstraps, the identity of these conditions was then randomly switched to obtain a null distribution of paired firing rate differences. We then identified where the actual firing rate difference lays on the bootstrapped distribution and recorded the percentile to calculate a two-tailed $p$-value. We chose more control markers to increase the reliability of baseline estimation. Finally, $p$-values were adjusted by the false-discovery rate (function fdr_bh.m[70]).

Mean firing rates during the down-state were computed between the two zero-crossings of the mean LoWS (from −0.15 to 0.165 s around the LoWS trough) and normalized by the mean firing rate during the surrounding baseline window (−2.9 to −0.2 and 0.2 to 2.9 s around the LoWS trough). For statistical analysis, these values were compared against the firing rate occurring around nSW and pSW, which occurrence had been shuffled (left end of each line in Fig. 2e).

## Analysis of IEDs, LoWS and memory performance

To assess the effect of IEDs and LoWS on memory performance, we tested whether mean incidence rates during encoding (i.e., during the 6 s of image presentation) and retrieval (i.e., during the variable duration period between the associative memory question appearing and participant's response) differed between high and low accuracy trials (i.e., those with correct versus incorrect responses) and fast vs slow RT trials. To assess the effect on RTs, we computed a linear mixed model using RT as the dependent variable and IED rate and SW rate during encoding and retrieval as independent variables.

## Statistical analyses

Using the web application estimationstats.com[64], we performed estimation statistics rather than significance testing for most statistical analyses. Estimation statistics provide numerous advantages[64], one of them being that the statistical analyses relies on an evaluation of the effect size. For these analyses, the $p$-values displayed represent the probability of obtaining the same effect size if the mean paired difference is equal to 0. We indicate the mean paired difference (effect size, ES) as well as the 95% confidence interval (95% CI). $P$-values are obtained through two-sided permutation $t$-tests using 5000 bootstraps. Since estimation statistics are not null-hypothesis testing, they do not require correction for multiple comparisons. Even so, we checked whether $p$-values survived a Bonferroni–Holm correction when multiple comparisons were applied: % of unit spiking activity showing a decrease during down-state; LoWS effect on behavior; IEDs effect on behavior; LoWS slope during encoding and retrieval; LoWS amplitude during encoding and retrieval; LoWS rate during encoding and retrieval. We found that all $p$-values remained significant. When multiple comparisons were applied on null-hypothesis statistics, we also applied a Bonferroni–Holm correction. For most plots, we display the effect size on the right of the respective panel.

## Reporting summary

Further information on research design is available in the Nature Portfolio Reporting Summary linked to this article.

## Data availability

This work is based on clinical data that was subject to ethics committee approval and patient consent. We will share clinical data on request, provided that request fulfils the ethics approval that we have for data collection and analysis. Source data for the main Results are provided with the paper. The MNI dataset is accessible on https://mni-open-ieegatlas.research.mcgill.ca/. The dataset is powered by LORIS which is under license GPLv3. The Boran dataset is accessible on https://doi.gin.g-node.org/10.12751/g-node.d76994/. The Boran dataset is under license CC BY-SA 4.0 DEED. Source data are provided with this paper.

## Code availability

Code for slow wave and interictal epileptiform discharge detection is available at https://github.com/bushlab-ucl/slowWaveDetection. Frequency analyses were performed using Fieldtrip (https://www.fieldtriptoolbox.org/). Single and multi-unit activity were identified using wave_clus (version 3.0.3, https://github.com/csn-le/wave_clus). We also used Matlab (v. 2022a), Graphpad Prism (v. 9 and updates, also for illustration), Adobe Illustrator (v. 26 and updates, for illustration only), SPSS (v. 29) and the web application EstimationStats (https://www.estimationstats.com/#/).

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

## Acknowledgements

We thank Dimitri Kullmann for critically reviewing the manuscript. We thank Ece Boran, Johannes Sarnthein, Birgit Frauscher and all the contributors of the MNI Open iEEG Atlas for sharing their respective datasets. This work was supported by the Department of Health's National Institute for Health Research, UCL/UCL Biomedical Research Centre. L.S. is funded by Swiss National Science Foundation P500PM_206720. U.V. is funded by Academy of Medical Sciences, ER UK, MRC (MR/T033150/1), Medical Research Foundation and NIHR BRC. D.B. is funded by a UKRI Frontier Research Grant (EP/X023060/1). F.A.C. is funded by MRC CARP. B.D. is funded by ER UK. N.B. is funded by Wellcome Principal Research Fellowship (222457/Z/21/Z). RR is funded by Wellcome Trust Innovation Program (218380/Z/19/Z).

## Author contributions

Designed the study: L.S., U.V., D.B., N.B., M.C.W. Data acquisition: U.V., R.R., B.D., F.A.C., A.W.M., A.M., J.A.B., D.B. Analyzed the data: L.S., U.V., R.R., D.B., N.B., M.C.W. Wrote first draft of the manuscript: L.S., U.V., D.B., N.B., M.C.W. All authors read and commented on subsequent drafts

## Competing interests

The authors declare no competing interests.
