## [Peer Review File · Nature Communications]

Wake slow waves in focal human epilepsy impact network activity and cognitionEditorial Note: This manuscript has been previously reviewed at another journal that is not operating a transparent peer review scheme. This document only contains reviewer comments and rebuttal letters for versions considered at *Nature Communications*.

REVIEWER COMMENTS

Reviewer #1 (Remarks to the Author):

The authors have appropriately addressed many of the comments I made in the first review. Working with the response document was however not easy because the authors did not, most often, refer to the specific place of the manuscript where they made a change (e.g. “a change was made in the Discussion”), and did not replicate the text of the change in the response document.

The authors will also see that I raised new points that were not raised in the first review. I hope they will appreciate that with a manuscript of this complexity and the months elapsed between the first and second review, it is possible to notice the second time issues that were missed the first time but are thought of importance.

The abstract states the hypothesis that “slow waves are activated in epilepsy to reduce aberrant activity”. On one hand the authors claim that the local wake slow waves (LoWS) are a normal phenomenon, but here that say that they are “activated in epilepsy”. I do not think they have demonstrated that LoSW are more frequent in epilepsy than in healthy brain or are in any way “activated”.

Comment 6. The issue remains that the rarity of wake slow wave is not a result but is a necessary consequence of the definition (top 10%). If the denominator is the number of slow waves detected during the wake period analyzed, they will necessarily find that LoSW are rare. They could not find, for instance, a frequency of 0.4/s/channel. They find this type of rate in N3 of the MNI dataset because I assume they took the top 10% of all slow waves recorded during the different stages of sleep. If they normalized separately for each state, they would find that slow waves are rare in all sleep stages, including N3. It must be clear that rarity of wake slow waves is not a finding of this study, which studies the properties of a small selection of slow waves, the largest ones. The first section of the results should be modified and the influence of the definition of slow waves must be discussed.

Comment 11. It appears that no explanation was placed in the manuscript. This is required.

Comment 12. Then the conclusions should clearly be limited to the temporal lobe. In the MNI dataset, as in this dataset, the LoSW are rare by construction. The authors can conclude that they are less frequent in wake than in N3 from the MNI data, but in fact they are not also not frequent in N2. See comment 6 above.

Comment 13. I was asking for the high pass filter, not the low pass, which is much less important in the context of this study. If the high-pass is too high, it could interfere with the detection of slow waves and this should be discussed.

Comment 14. The authors did not explain how they assessed that the patients were awake. If there was a direct assessment, it should be explained. If there was no direct assessment, then they should say that the assessment was done by comparing parts of the spectrum between the task period and the presumably awake period. The absence of a statistically significant difference (particularly with a $p=0.07$) is of course not a proof of similarity. This is an important issue because the authors describe an unusual phenomenon which, they claim, occurs during wakefulness. One must know how this was assessed and the possible uncertainties in this process.

Comment 15. Information about the reference is still not given, even in the section called "choice of the reference". Where is the reference?

Comment 16. The manuscript was not modified to reflect the authors' answer. If the visual override included spikes within 1 s of an automatic detection, what is the purpose of this 1s refractory period? Is it meaningless?

Comment 20: the authors can keep these sections but should make it clear that the absence of correlation only indicates that the majority of LoSW are not IEDs, not excluding the possibility that some LoSW are mislabeled IEDs.

Results around line 237. The increase of slope and amplitude of the LoSW are so small that it is important to make sure they are not just correlated to passing time (since last sleep, until next sleep). It would be easy and useful to place randomly fake LoSW at the same average rate as LoSW and verify that the changes are then absent. The authors argue that they occur at a rate similar to changes occurring during a night. This is indeed a very small change if considered over 1 to 3 min. Does it have a practical effect?

With respect to the change in gamma power as a function of duration since the last LoSW, it is difficult to appreciate its magnitude. Could it be given, like the slope, in %/min? Could the

authors comment on the functional significance of this magnitude, i.e. does this change have a chance to visibly alter the number or “size” of IEDs?

Reviewer #2 (Remarks to the Author):

In this manuscript, Sheybani et al. study the role of slow waves on network activity and cognition during wakefulness in patients with focal epilepsy. They found that high gamma and neural spiking decreases during these LoWS. The waveform features of LoWS change over time leading up to the next IED. A longer delay since the last LoWS is associated with higher network excitability during IEDs. LoWS properties are predominantly altered during memory encoding, and higher occurrence of LoWS is associated with slower reaction time, but no effect on accuracy, of a visual association task. The authors hypothesize that there could be a homeostatic role for LoWS and posit that this work could be used as the foundation for interventional strategies in epilepsy patients.

The authors have included additional methodological detail, as well as incorporated data from a new database to strengthen their claims.

- 1) The additional analyses presented to clarify the difference between LoWS and lesional focal slowing are appreciated. The interpretation of these findings would benefit from further measured discussion. It is still unclear to me whether the authors posit that LoWS are induced by the abnormal network dynamics that are a consequence of epilepsy, or if they are a physiologic phenomenon that also occurs in epilepsy. The wording “healthy brain regions” and “not restricted to epileptic brain regions” (pg. 5, line 128-130) seems to argue for the latter. Do LoWS still play a homeostatic role in this case? The assignment of LoWS as key components of epilepsy homeostasis (pg 10, line 321) is more in keeping with the former. I understand it is a complex issue, but this is a central point of the manuscript.
- 2) Although mechanistic experimentation is beyond the scope of the manuscript, some discussion of how the authors hypothesize that LoWS induce relatively long-lasting changes in network excitability should be included. For instance, it is conceivable that the occurrence of a LoWS could actually be hypersynchronizing by causing a strong reduction in firing rate across the population simultaneously. How would this decrease the likelihood of

subsequent hypersynchronous phenomena (IEDs)? Such a discussion would be helpful to frame future mechanistic studies to understand the authors' observed correlations.

3) The 'n' used for each analysis should be presented somewhere (either in the Results text or figure legends, but in a consistent manner). Can the authors confirm in the manuscript that the linear mixed models used incorporate patients as a grouping variable to ensure that datapoints from a restricted subset of patients are not driving the overall calculated effect size.

We are very grateful for the reviewers careful and thorough reviews of our manuscript. We have addressed all the points raised and feel that this has substantially improved our study.

General comments:

- a) Subheadings of the Results have been updated to comply with the journal's formatting instructions.
 - Identification of slow waves during wakefulness in patients with epilepsy (old)
➔ Slow waves during wakefulness in patients with epilepsy (new)
 - Negative slow waves during wakefulness share key characteristics with sleep slow waves (old)
➔ Wake slow waves share key characteristics with sleep slow waves (new)
 - LoWS are associated with a down-state of neuronal firing, capturing a core feature of sleep slow waves (old)
➔ LoWS are associated with a down-state of neuronal firing (new)

- b) We removed some citations to comply with the journal's formatting instructions:

Contreras, D. & Steriade, M. Cellular basis of EEG slow rhythms: a study of dynamic corticothalamic relationships. *The Journal of Neuroscience* 15, 604–622 (1995)

Steriade, M., Nuiiez, A. & Amzica, F. A Novel Slow (<1 Hz) Oscillation of Neocortical Neurons in viva: Depolarizing and Hyperpolarizing Components. *The Journal of Neuroscience* 14 (1993)

Claridge-Chang, A. & Assam, P. N. Estimation statistics should replace significance testing. *Nat Methods* 13, 108–109 (2016)

Dijk, D.-J. Regulation and Functional Correlates of Slow Wave Sleep. *Journal of Clinical Sleep Medicine* 5, (2009)

Steriade, M. *Neuronal Substrates of Sleep and Epilepsy*. (2004)

- c) Figures' legends have been adapted to comply with the journal's formatting instructions
- d) Colors in the figures have been adapted to comply with the journal's formatting instructions
- e) The error reported in the linear mixed models are s.e.m, not s.d

Reviewer #1 (Remarks to the Author):

The authors have appropriately addressed many of the comments I made in the first review. Working with the response document was however not easy because the authors did not, most often, refer to the specific place of the manuscript where they made a change (e.g. "a change was made in the Discussion"), and did not replicate the text of the change in the response document.

We apologize for the lack of clarity. In the current response to reviewers, we provide where in the manuscript each point-by-point response is addressed.

The authors will also see that I raised new points that were not raised in the first review. I hope they will appreciate that with a manuscript of this complexity and the months elapsed between the first and second review, it is possible to notice the second time issues that were missed the first time but are thought of importance.

1. The abstract states the hypothesis that “slow waves are activated in epilepsy to reduce aberrant activity”. On one hand the authors claim that the local wake slow waves (LoWS) are a normal phenomenon, but here that say that they are “activated in epilepsy”. I do not think they have demonstrated that LoSW are more frequent in epilepsy than in healthy brain or are in any way “activated”.

We agree with R1 that this might be an overstatement. We have removed it from the manuscript. It reads:

Previous version	Current version	page (version with track-changes)
that slow waves are activated in epilepsy to reduce aberrant activity (...)	that slow waves are activated in epilepsy to reduce aberrant activity (...)	Abstract, p. 1
it was previously unknown whether such beneficial activity could also be activated during wakefulness in pathological conditions	it was previously unknown whether such beneficial activity could also be at play activated during wakefulness in pathological conditions	Discussion, p. 11
To summarize, our study provides evidence of “micro-sleep” modules activated during wakefulness in epilepsy (...)	To summarize, our study provides evidence of “micro-sleep” modules activated during wakefulness in epilepsy (...)	Discussion, p. 14

2. Comment 6. The issue remains that the rarity of wake slow wave is not a result but is a necessary consequence of the definition (top 10%). If the denominator is the number of slow waves detected during the wake period analyzed, they will necessarily find that LoSW are rare. They could not find, for instance, a frequency of 0.4/s/channel. They find this type of rate in N3 of the MNI dataset because I assume they took the top 10% of all slow waves recorded during the different stages of sleep. If they normalized separately for each state, they would find that slow waves are rare in all sleep stages, including N3. It must be clear that rarity of wake slow waves is not a finding of this study, which studies the properties of a small selection of slow waves, the largest ones. The first section of the results should be modified and the influence of the definition of slow waves must be discussed.

It is true that the rate of an event such as slow waves will be influenced by the threshold that is used for detection, whether determined by an absolute threshold (as in Massimini et al. (2004) *J Neurosci*) or a relative threshold (as in Frauscher et al. (2015) *Brain*). We thus now limit our claim to the fact that SW during wakefulness are rarer than during sleep.

We had removed from the initial submission two specific remarks in which we mentioned the rarity of SW. This might have been overlooked, as we did not highlight it in our previous Response to Reviewer. However, these claims regarding the rarity of SW have been removed:

- “[the rate of co-occurrence across the hippocampus, amygdala and temporal neocortex was] remarkably close to the percentage observed for sleep SW” (p. 5, lines 146-147 of the initial submission)
- “nSW occurred at a mean \pm SD rate of 0.06/s/electrode \pm 0.02 (Extended Data Fig. 1a), well below the rate described during sleep” (p. 4, lines 112-113 of the initial submission)

We have added a note in the Discussion regarding this point:

Previous version	Current version	page (version with track-changes)
n/a	Nonetheless, our results are based on wake SW with a particularly high amplitude, detected by an algorithm like that used by Frauscher et al. (2015). While it is not clear if the effects we observe therefore generalise to all (lower amplitude) SWs, this strategy was necessary to ensure that identified events were unlikely to be noise.	Discussion, p. 12

3. Comment 11. It appears that no explanation was placed in the manuscript. This is required.

Apologies for this omission. The information is now provided in the Methods:

Previous version	Current version	page (version with track-changes)
n/a	Patients with microelectrode recordings also had macroelectrodes implanted, but there were technical issues with the synchronization of both signals which prevented us from including those macroelectrode data.	Methods, p. 14

4. Comment 12. Then the conclusions should clearly be limited to the temporal lobe. In the MNI dataset, as in this dataset, the LoSW are rare by construction. The authors can conclude that they are less frequent in wake than in N3 from the MNI data, but in fact they are not also not frequent in N2. See comment 6 above.

The fact that we worked on temporal lobe electrodes is indicated in the abstract (“Using intracortical recordings from the temporal lobe of 25 patients (...”). We have also improved the introduction as follows:

Previous version	Current version	page (version with track-changes)
Together, our findings indicate that LoWS with key features of sleep SW dynamically respond (...)	Together, our findings indicate that temporal lobe LoWS with key features of sleep SW dynamically respond (...)	Introduction, p. 3

We have indeed limited our interpretation, regarding the rate of LoWS, to the fact that they are less frequent than during sleep (see our response to the 2nd comment of the current Response to Reviewer).

We have added a note regarding the observation that they are less frequent during N2 than N3:

Previous version	Current version	page (version with track-changes)
n/a	The lower rate during NREM 2 probably reflects the fact that, by definition (Fiorillo et al., 2019), a maximum of 20% of slow-wave activity can be detected in any 30 s window, while NREM 3 can be comprised of 20-100% of slow-wave activity.	Results, p. 5

5. Comment 13. I was asking for the high pass filter, not the low pass, which is much less important in the context of this study. If the high-pass is too high, it could interfere with the detection of slow waves and this should be discussed.

The high-pass was 0.15 Hz. This has been added to the Methods:

Previous version	Current version	page (version with track-changes)
(...) with a low-pass hardware filter set at a cut-frequency equal to ¼ of the sampling rate, using a Micromed (...)	(...) with a low-pass hardware filter set at a cut-frequency equal to ¼ of the sampling rate and a 0.15 Hz high-pass, using a Micromed (...)	Methods, p. 15

6. Comment 14. The authors did not explain how they assessed that the patients were awake. If there was a direct assessment, it should be explained. If there was no direct assessment, then they should say that the assessment was done by comparing parts of the spectrum between the task period and the presumably awake period. The absence of a statistically significant difference (particularly with a $p=0.07$) is of course not a proof of similarity. This an important issue because the authors describe an unusual phenomenon which, they claim, occurs during wakefulness. One must know how this was assessed and the possible uncertainties in this process.

One of the researchers was present with each subject during the cognitive task period. They confirmed that subjects were responsive and vigilant. Between the task periods, patients were seen intermittently and no sleep was noted. Although the lack of difference is not proof of similarity, the fact that delta and theta power tend to be lower during rest periods, in comparison to cognitive task periods, is a strong argument that subjects were not sleeping. The trend for a higher power of delta/theta might be related to memory processing (see e.g. Lega et al., 2012, *Hippocampus*). Two important criteria to score NREM 1 are: (a) posterior alpha rhythm disappears and is replaced by theta rhythm; (b) peak frequency of posterior alpha rhythm decreases by 1 Hz. Supplementary Fig. 10a confirms that posterior alpha power does not decrease. Furthermore, our new analysis in Supplementary Fig. 10b confirms that alpha peak frequency is not lower during rest periods. Hence, there is no evidence for sleep between the cognitive task periods. Finally, we also note that LoWS actually increased during the cognitive task.

We have updated the methods and Supplementary Fig. 10 accordingly.

Methods:

	Previous version	Current version	page (version with track-changes)
1.	Patients were alert during recording.	Patients were alert and responsive during cognitive task periods recording, as assessed directly by one of the co-authors (UV, JB, or DB) in the recording room. Between cognitive task periods, patients were seen intermittently, and no evidence of sleep was noted. Furthermore, since some recordings (...)	Methods, p. 16
2.	2 way ANOVA, effect of condition [during vs	2 way ANOVA, effect of condition [during vs outside task]: $F(1, 16)=3.83, p=0.07$; interaction condition *	Methods, p. 16

	outside task]: $F(1, 16)=3.83, p=0.07$, Supplementary Fig. 10	frequency: $F(1.685, 26.97) = 0.1334, p=0.8416$, Supplementary Fig. 10a	
3.	n/a	It is possible that the trend for an increase in delta power during the cognitive task reflects the increased incidence of LoWS (although, as mentioned earlier, there was no correlation between delta power and LoWS rate, Supplementary Fig. 3a), or could be a consequence of memory processing (Lega et al., 2012). A standard criterion of NREM 1 is a decrease in alpha peak frequency by 1 Hz (Troester et al., 2023); but in our data, alpha peak frequency was actually significantly higher during rest periods than during the cognitive task (during rest, mean \pm SD: 9.74 Hz \pm 0.6 Hz; during cognitive task: 9.61 Hz \pm 0.58 Hz, paired t-test, $p=0.0262$, Supplementary Fig. 10b). Hence, the rest period was not marked by two fundamental properties of NREM 1 (a decrease in alpha power and peak frequency), and delta/theta activity also tended to be higher during rest periods, implying that patients did not sleep between cognitive task periods.	Methods, p. 16

The caption of Supplementary Fig. 10 has been updated:

	Previous version	Current version	page (version with track-changes)
1.	No change in power between the task and the rest period in-between	Rest and cognitive task periods present the same spectral signatures No change in power between the task and the rest period in-between	Supplementary Fig. 10
2.	n/a	(b) Peak alpha frequency is slightly higher during rest (9.7 Hz) than during the cognitive task (9.6 Hz).	Supplementary Fig. 10

7. Comment 15. Information about the reference is still not given, even in the section called “choice of the reference”. Where is the reference?

The reference was consistently located in white matter, remote from the epileptic focus, although the exact cortical location differed across patients. This information is now provided in the Methods:

Previous version	Current version	page (version with track-changes)
Macroelectrode EEG data were recorded continuously at a sample rate of (...)	Macroelectrode EEG data were recorded continuously against a common reference located in white matter remote from the epileptic focus at a sample rate of (...)	Methods, p. 15

8. Comment 16. The manuscript was not modified to reflect the authors' answer. If the visual override included spikes within 1 s of an automatic detection, what is the purpose of this 1s refractory period? Is it meaningless?

We apologize for this omission. As in practice this refractory period was indeed meaningless, we removed it from the manuscript to avoid any confusion.

Previous version	Current version	page (version with track-changes)
A refractory period of 1 s after each candidate IED was added, during which no other IED could be detected.	A refractory period of 1 s after each candidate IED was added, during which no other IED could be detected.	Methods, p. 17

9. Comment 20: the authors can keep these sections but should make it clear that the absence of correlation only indicates that the majority of LoSW are not IEDs, not excluding the possibility that some LoSW are mislabeled IEDs.

We have updated the manuscript accordingly:

Previous version	Current version	page (version with track-changes)
n/a	To note, steps 4 and 5 only indicate that the majority of LoWS are not IEDs, but the convergence of all 8 findings strongly supports the hypothesis that LoWS are not mislabelled IEDs.	Methods, p. 20

10. Results around line 237. The increase of slope and amplitude of the LoSW are so small that it is important to make sure they are not just correlated to passing time (since last sleep, until next sleep). It would be easy and useful to place randomly fake LoSW at the same average rate as LoSW and verify that the changes are then absent.

We thank R1 for this very important remark. As a way to control for this, we included the delay since the start of recording as a random variable and still observed a significant association between slope or amplitude and delay to the next IED.

We have thus updated:

	Previous version	Current version	page (version with track-changes)
1.	(...) we found that the slope (β -estimate \pm SD: slope increases by 1% each 371 s \pm 57, adj. $p=1\cdot 10^{-8}$, $n=16$, Fig. 5a, b) and amplitude (β -estimate \pm SD: amplitude increases by 1% each 174 s \pm 49, adj. $p=5\cdot 10^{-4}$, $n=16$, Fig. 5a, c)	(...) we found that the slope (β -estimate \pm sem SD: slope increases by 0.001% 1% each 1 s 371 \pm 57, adj. $p=4\cdot 10^{-17}$ 1 $\cdot 10^{-8}$, $n=16$, Fig. 5a, b) and amplitude (β -estimate \pm sem SD: amplitude increases by 0.002% 1% each 1 s 174 s \pm 49, adj. $p=8\cdot 10^{-58}$ 5 $\cdot 10^{-4}$, $n=16$, Fig. 5a, c)	Results, p. 8

2.	LoWS-slope before IEDs. To measure variability in the slope of LoWS before IEDs, we computed the average slope of all LoWS in each 5 s time window relative to the closest IED (up to 250 s before IEDs). To do so, we identified the minimum of the 0.5-4 Hz filtered signal in a 1 s window following LoWS onset and defined LoWS slope as the ratio between the amplitude and post-onset time of this minima. Each slope was normalized by the value of the highest absolute slope for that patient, so that all values lay in the range of 0 to 1. These values and amplitude of LoWS (see below) were then used as covariates in the linear mixed model. Amplitude of LoWS before IEDs. We followed the same procedure described above but using the absolute minima of the 0.5-4 Hz filtered signal in a 1 s window following LoWS onset, in place of the estimate of LoWS slope.	LoWS-slope and amplitude before IEDs. To measure variability in the slope of LoWS before IEDs, we computed the average slope of all LoWS in each 5 s time window relative to the closest IED (up to 250 s before IEDs). To do so, To look for changes in the slope and amplitude of LoWS leading up to IEDs, we first identified the minimum of the 0.5-4 Hz filtered signal in a 1 s window following LoWS onset (which was defined as the amplitude) and defined LoWS slope as the ratio between that amplitude and the post-onset time of this minima. Slope and amplitude values were then normalized by the highest absolute value of the highest absolute slope and amplitude observed in each for that patient, so that all values lay in the range of 0 to 1. These values and amplitude of LoWS (see below) were then used as dependent variables covariates in separate linear mixed models. Time to the next IED was included as a fixed factor. We also included time since the start of recording as a random factor, to account for the possibility that any observed changes in slope and amplitude are simply due to the accumulation of sleep pressure over the course of testing. Finally, we included patient identity as a grouping variable, in this and all other linear mixed model analyses. For display purposes, panels b and c of Fig. 5 were constructed from the average slope and amplitude respectively by bins of 5 s. Amplitude of LoWS before IEDs. We followed the same procedure described above but using the absolute minima of the 0.5-4 Hz filtered signal in a 1 s window following LoWS onset, in place of the estimate of LoWS slope.	Methods, p. 21-22
----	---	--	-------------------

Figure 5 has been updated accordingly.

- The authors argue that they occur at a rate similar to changes occurring during a night. This is indeed a very small change if considered over 1 to 3 min. Does it have a practical effect?

This is a good point indeed. Our interpretation is based on several findings (the increase in HG before IEDs, the increase of LoWS slope and amplitude, and the increase of IED-HG with time since the last LoWS) which suggest that LoWS decrease the excitability of IEDs. This also relates to comment #2 from Reviewer 2. We have now included further discussion of this point:

Previous version	Current version	page (version with track-changes)
n/a	What could be the underlying mechanism for a protective effect of LoWS? Evidence suggests that SW are instrumental in decreasing neuronal excitability during sleep (Vyazovskiy et al., 2008, 2009). In particular, sleep SW orchestrate increased neuronal synchronization (i.e., low frequency bursts of population activity) that has been shown to promote synaptic downscaling (i.e., a decrease in synaptic strength) (Czarnecki et al., 2007). Since neuronal synchronization has been shown to correlate with sleep SW slope (Vyazovskiy et al., 2009), the increase in LoWS slope that we observed before IEDs (although relatively small across this much shorter timescale) might also reflect increased neuronal synchronization, which promotes synaptic downscaling and could therefore underlie the reduction in HG power during subsequent IEDs. Further studies, including manipulation protocols, will be necessary to demonstrate causality.	Discussion, p. 11-12

12. With respect to the change in gamma power as a function of duration since the last LoSW, it is difficult to appreciate its magnitude. Could it be given, like the slope, in %/min? Could the authors comment on the functional significance of this magnitude, i.e. does this change have a chance to visibly alter the number or “size” of IEDs?

We thank R1 for this important comment. We now provide a magnitude of the change in HG in a similar way as we did for slope and amplitude. To provide a more intuitive value, we removed a normalization step, which did not qualitatively affect the results. We can now demonstrate that this change ($\sim 1 \mu V^2$ per s) corresponds to an increase of $\sim 0.9\%$ of the mean HG power across all recordings, with every additional 1 s that passes since the last LoWS. The modifications are as follow:

	Previous version	Current version	page (version with track-changes)
1.	We found a significant effect of the delay since the last LoWS on HG power during IEDs (β -estimate \pm SD: $7.8 \cdot 10^{-5} \pm 1.2 \cdot 10^{-5} \log(\mu V^2 / \mu V^2) / s$, $p=3 \cdot 10^{-11}$, Fig. 5d). This indicates that an increased delay since the last LoWS is associated with a relative increase of HG during the IED, suggesting that any beneficial impact (...)	We found a significant effect of the delay since the last LoWS on HG power during IEDs (β -estimate \pm sem SD: $2 \cdot 10^{-3} \log(\mu V^2) s^{-1} \pm 3 \cdot 10^{-4}$, $n=16$, $p=1 \cdot 10^{-10}$, $7.8 \cdot 10^{-5} \pm 1.2 \cdot 10^{-5} \log(\mu V^2 / \mu V^2) / s$, $p=3 \cdot 10^{-11}$, Fig. 5d), indicating that for every additional 1 s since the last LoWS, HG power during IEDs increases by $\sim 1 \mu V^2$, or 0.9% of the grand average HG power across patients (mean \pm SD: $113.1 \mu V^2 \pm 56.7$, $n=17$ patients). This indicates that an increased delay since the last LoWS is associated with a relative increase of HG during the	Results, p. 9

		IED -This suggests that any beneficial impact (...)	
2.	We measured the HG power during IEDs (-0.05 to 0.05 s around IEDs) and normalized it against HG power sampled randomly during artefact-free periods. Then, we used a linear mixed model with HG power during IED as the dependent variable and delay since the last LoWS as a covariate.	We measured the HG power during IEDs (-0.05 to 0.05 s around IEDs) and normalized it against HG power sampled randomly during artefact-free periods. Then, we was used in a linear mixed model with HG power during IED as the dependent variable and delay since the last LoWS as the independent variable a covariate	Methods, p. 23
3.	n/a	Mean HG power across recordings was measured using the same parameters, randomly sampling the recordings with windows of similar size (500 bootstraps).	Methods, p. 23

Reviewer #2 (Remarks to the Author):

In this manuscript, Sheybani et al. study the role of slow waves on network activity and cognition during wakefulness in patients with focal epilepsy. They found that high gamma and neural spiking decreases during these LoWS. The waveform features of LoWS change over time leading up to the next IED. A longer delay since the last LoWS is associated with higher network excitability during IEDs. LoWS properties are predominantly altered during memory encoding, and higher occurrence of LoWS is associated with slower reaction time, but no effect on accuracy, of a visual association task. The authors hypothesize that there could be a homeostatic role for LoWS and posit that this work could be used as the foundation for interventional strategies in epilepsy patients.

The authors have included additional methodological detail, as well as incorporated data from a new database to strengthen their claims.

1) The additional analyses presented to clarify the difference between LoWS and lesional focal slowing are appreciated. The interpretation of these findings would benefit from further measured discussion. It is still unclear to me whether the authors posit that LoWS are induced by the abnormal network dynamics that are a consequence of epilepsy, or if they are a physiologic phenomenon that also occurs in epilepsy. The wording “healthy brain regions” and “not restricted to epileptic brain regions” (pg. 5, line 128-130) seems to argue for the latter. Do LoWS still play a homeostatic role in this case? The assignment of LoWS as key components of epilepsy homeostasis (pg 10, line 321) is more in keeping with the former. I understand it is a complex issue, but this is a central point of the manuscript.

We agree with R2 that this is a complex and important question. As inferred by R2, the presence of LoWS in the MNI dataset and the fact that they do not co-localize with the epileptic focus suggest that they are, at least in part, physiological activities. As sleep SW, we expect them to decrease neuronal excitability, including that involved in epileptiform discharges. We have added this interpretation in the Discussion:

Previous version	Current version	page
------------------	-----------------	------

		(version with track-changes)
n/a	While it is difficult to infer whether LoWS are physiological or induced by the presence of epilepsy, we do not expect them to be entirely specific to epilepsy. Indeed, we were able to identify LoWS in presumably healthy brain regions within the MNI dataset; and furthermore, they do not show evidence of co-localization with the epileptic focus. Hence, our interpretation is that they represent, at least in part, physiological activities that impact on neuronal excitability, including that involved in epileptiform discharges.	Discussion, p. 11

2) Although mechanistic experimentation is beyond the scope of the manuscript, some discussion of how the authors hypothesize that LoWS induce relatively long-lasting changes in network excitability should be included. For instance, it is conceivable that the occurrence of a LoWS could actually be hypersynchronizing by causing a strong reduction in firing rate across the population simultaneously. How would this decrease the likelihood of subsequent hypersynchronous phenomena (IEDs)? Such a discussion would be helpful to frame future mechanistic studies to understand the authors' observed correlations.

We share R2's view on the potential underlying mechanism explaining the association between delay since last LoWS and decreased HG during IEDs. SW increase neuronal synchronization, which has been shown to favour synaptic downscaling. We hypothesize that a similar mechanism might be at play in the context of epilepsy. This also relates to R1's comment #11. We have added this paragraph in the Discussion (as indicated in R1's comment #11):

Previous version	Current version	page (version with track-changes)
n/a	What could be the underlying mechanism for a protective effect of LoWS? Evidence suggests that SW are instrumental in decreasing neuronal excitability during sleep (Vyazovskiy et al., 2008, 2009). In particular, sleep SW orchestrate increased neuronal synchronization (i.e., low frequency bursts of population activity) that has been shown to promote synaptic downscaling (i.e., a decrease in synaptic strength) (Czarnecki et al., 2007). Since neuronal synchronization has been shown to correlate with sleep SW slope (Vyazovskiy et al., 2009), the increase in LoWS slope that we observed before IEDs (although relatively small across this much shorter timescale) might also reflect increased neuronal synchronization, which promotes synaptic downscaling and could therefore underlie the reduction in HG power during subsequent IEDs. Further studies, including manipulation protocols, will be necessary to demonstrate causality.	Discussion, p. 11-12

3) The 'n' used for each analysis should be presented somewhere (either in the Results text or figure legends, but in a consistent manner). Can the authors confirm in the manuscript that the linear mixed

models used incorporate patients as a grouping variable to ensure that datapoints from a restricted subset of patients are not driving the overall calculated effect size.

We have now added the *n* used for each analysis throughout the Results, where previously omitted. In doing so, we noticed a minor error in our previous reporting of the results from the linear mixed model that examined the association between LoWS rate and reaction times (which does not qualitatively affect the result or change the interpretation). Regarding the linear mixed model analyses, we can confirm that patients were included as a grouping variable in all cases. We have added this information in the Methods:

Previous version	Current version	page (version with track-changes)
n/a	Importantly, the rate of LoWS/IEDs during retrieval was correlated with the accuracy of memory retrieval in that trial, whilst the rate of LoWS/IEDs during encoding was correlated with the accuracy of memory retrieval across both retrieval trials for the pair being presented on screen. This leads to a small difference in the number of patients included in each analysis.	Methods, p. 15
n/a	Finally, we included patient identity as a grouping variable, in this and all other linear mixed model analyses	Methods, p. 22

REVIEWER COMMENTS

Reviewer #1 (Remarks to the Author):

The authors have addressed thoroughly my comments and I am satisfied with their answers.

Reviewer #2 (Remarks to the Author):

The authors have adequately addressed my remaining points.